# Neural Networks Trained by Weight Permutation are Universal Approximators

## Abstract

The universal approximation property is fundamental to the success of neural networks, and has traditionally been achieved by networks without any constraints on their parameters. However, recent experimental research proposed an innovative permutation-based training method, which can achieve desired classification performance without modifying the exact values of the weights. In this paper, we prove that the permutation training method can guide a ReLU network to approximate one-dimensional continuous functions. Our numerical results under more diverse scenarios also validate the effectiveness of the permutation training method in regression tasks. Moreover, the notable observations during weight permutation suggest that permutation training can provide a novel tool for describing network learning behavior.

## 1 Introduction

The universal approximation property (UAP) of neural networks is a cornerstone in the theoretical guarantee of deep learning, proving that even the simplest two-layer feedforward networks can accurately approximate any continuous function (Cybenko, 1989; Hornik et al., 1989; Leshno et al., 1993). This fascinating ability allows neural networks to replace critical, challenging components in existing frameworks to enhance efficiency (Mnih et al., 2013; Goodfellow et al., 2014; Raissi et al., 2019). Despite the extensive study in various settings, existing research on the UAP rarely imposes restrictions on the network parameters. However, in certain application scenarios, constraints posed by some specific requirements are essential (Nugent, 2005; Kosuge et al., 2021a).

As a constrained scenario, Qiu & Suda (2020) empirically showed that, without altering the exact value of network weights, only permuting the initialized weight vector can achieve comparable or better performance for image classification tasks. This unique property makes the permutation-based training method attractive for specific hardware applications, such as fixed-weight accelerator (Kosuge et al., 2021a). It can also facilitate the implementation of physical neural networks (Nugent, 2005; Feldmann et al., 2021) (see App. A for details). Motivated by this impressive result, we are intrigued to investigate whether this permutation training method still possesses UAP.

In this paper, we theoretically establish the UAP of a permutation-trained network with rectified linear unit (ReLU) activation functions for any one-dimensional continuous function. To address the non-trivial challenge posed by the permutation training setting, the key idea of our proof is a four-pair construction of the step function approximators, which helps us to approach the targeted continuous function with a piecewise constant function (Stein & Shakarchi, 2009). Additionally, a processing method is proposed to eliminate the impact of the remaining weights.

Moreover, our numerical experiments not only validate the theoretical results but also demonstrate the widespread existence of the UAP of permutation training in diverse initializations. The patterns observed during permutation training also highlight its potential in describing learning behavior, relating to topics like the pruning technique (Frankle & Carbin, 2019) and continual learning (Maltoni & Lomonaco, 2019; Zeng et al., 2019). We summarize the main findings of this paper below:

- We prove the UAP of permutation-trained networks with equidistant initialization and pairwise random initialization to one-dimensional continuous functions.
- We conduct numerical experiments of regression problems under various generalized settings, identifying the common occurrence of the UAP of permutation training.

- By observing the permutation patterns, we find that permutation training could potentially serve as a new approach to describe the detailed network learning behaviors.

**Related works.** The UAP has been extensively studied in various settings, leading to many efficient applications. It is well known that fully connected networks are universal approximators for continuous functions (Cybenko, 1989; Hornik et al., 1989; Leshno et al., 1993). Additionally, the UAP of continuous functional and operator are presented by Chen & Chen (1995), giving rise to the operator learning formalisms such as DeepONet (Lu et al., 2021). However, the traditional UAP is suited to wide networks where the weights are freely adjusted. Our configuration is focused on a specific approach that only allows permuting weights.

Permutation is crucial in deep learning and closely relates to permutation equivariant or invariant networks (Cohen & Welling, 2016), designed to learn from symmetrical data (Zaheer et al., 2017; Lee et al., 2019). It is also evident in graph-structured data which inherently exhibit permutation invariance (Maron et al., 2019; Satorras et al., 2021). However, these works mainly concern issues with intrinsic symmetry, while permutation training is not limited to these scenarios.

As for the weight permutation attempts, Qiu & Suda (2020) empirically proposed the first (to our knowledge) weight-permuted training method, which exhibits comparable classification performance and has been practically applied as a fixed-weight accelerator (Kosuge et al., 2021a;b). A further discussion about this method's advantages in hardware implementation is given in App. A. Our work provides theoretical guarantees of this method and considers some regression tasks numerically. Additionally, Scabini et al. (2022) improved the initialization by rewiring neurons from the perspective of computer networks, but the training methods are unchanged.

Permutation training is also closely related to the permutation symmetry and linear mode connectivity (LMC) (Frankle et al., 2020; Entezari et al., 2021). The LMC suggests that after a proper permutation, most SGD solutions under different initialization will fall in the same basin in the loss landscape. Similarly, our permutation training also seeks a permutation to effectively improve performance. Therefore, the search algorithm utilized in LMC (Jordan et al., 2023; Ainsworth et al., 2023) can serve as a reference for the permutation training algorithm, and vice versa. Moreover, it would be interesting to explore the LMC between different permutation training solutions.

**Outline.** We state the main result in Sect. 2, which includes ideas to derive the main result. In Sect. 3, we provide a detailed construction of the proof. The numerical results of permutation training are presented in Sect. 4, along with the observation of permutation behavior during the training process. Finally, the conclusion is provided in Sect. 5. All formal proof of the theorems is in the Appendix.

## 2 NOTATIONS AND MAIN RESULTS

### 2.1 NERUAL NETWORKS ARCHITECTURE

We start with a two-layer feed-forward ReLU network with $N$ hidden neurons in even numbers (*i.e.*, $N = 2n$). It has the form of a linear combination of ReLU basis functions (noted as $\mathrm{ReLU}(z) = \max\{z, 0\}$) as $f(x) = \sum_{i=1}^{N} a_i \mathrm{ReLU}(w_i \cdot x + b_i) + c$. Particularly, we focus on approximating one-dimensional functions, so all weights are scalars ($w_i, b_i, a_i, c \in \mathbb{R}$). Since ReLU activation is positively homogeneous (*i.e.*, $\mathrm{ReLU}(\gamma x) = \gamma \mathrm{ReLU}(x)$ for all $\gamma > 0$), we consider a simplified homogeneous case with $w_i = \pm 1$, and utilize $n$ to divide the basis functions into two parts as

$$\phi_i^{\pm}(x) = \mathrm{ReLU}\big(\pm (x - b_i)\big), \quad i = 1, 2, ..., n, \tag{1}$$

where the biases $\{b_i\}_{i=1}^{n}$ determine the location of basis functions. Then we introduce a one-dimensional linear layer. It will be shown later that while this layer is not essential for achieving UAP, it does simplify the proof and offer practical value. The network's output function $f^{\mathrm{NN}}$ gives

$$f^{\mathrm{NN}}(x) = \alpha + \gamma \sum_{i=1}^{n} \big[p_i \phi_i^{+}(x) + q_i \phi_i^{-}(x)\big], \tag{2}$$

where $\{p_i, q_i\}_{i=1}^{n}$ are the coefficients of the basis functions and $\alpha, \gamma$ are scaling factors. This form corresponds to a three-layer network, where $\{p_i, q_i\}_{i=1}^{n}$ and $\{\alpha, \gamma\}$ are the parameters in the second hidden layer and output layer, respectively.

## 2.2 Weight configuration and main theorems

Without loss of generality, we consider the target continuous function $f^* \in C([0,1])$. During the permutation training process, we hold the initial value of the second hidden layer's weight vector $\theta^{(n)} = (p_i, q_i)_{i=1}^n$ and only update the order relationship of its components, leading to the following configuration: the weight vector $\theta^{(n)}$ is permuted from a predetermined vector $W^{(n)} \in \mathbb{R}^{2n}$. We first focus on a simple scenario with equidistantly distributed location $B^{(n)}$ and pairwise coefficients $W^{(n)}$. The UAP of a permutation-trained network to continuous functions can be stated as follows:

**Theorem 1** (UAP with a linear layer). *For any function $f^*(x) \in C([0,1])$ and any small number $\varepsilon > 0$, there exists a large even number $n \in \mathbb{Z}^+$, and $\alpha, \gamma \in \mathbb{R}$ for $f^{NN}$ in Eq. (2) with equidistantly distributed $B^{(n)} = \left(0, \frac{1}{n-1}, \cdots, 1\right) =: (b_i)_{i=1}^n$ and corresponding $W^{(n)} = (\pm b_i)_{i=1}^n$, along with a permuted weight vector $\theta^{(n)} = \tau(W^{(n)})$, such that $|f^{NN}(x) - f^*(x)| \leq \varepsilon$ for all $x \in [0,1]$.*

The intuition of this result comes from the rich expressive possibility of permutation training since there are $(2n)!$ different permutations for $W^{(n)}$ [1]. Next, we enhance the result in Theorem 1 to a purely permuted situation, suggesting the UAP can be achieved without changing $\alpha, \gamma$ as

**Theorem 2** (UAP without the linear layer). *Let $\alpha = 0, \gamma = 1$. For any function $f^*(x) \in C([0,1])$ and any small number $\varepsilon > 0$, there exists a large even number $n \in \mathbb{Z}^+$ for $f^{NN}$ in Eq. (2) with equidistantly distributed $B^{(n)} = (b_i)_{i=1}^n$ and $W^{(n)} = (\pm b_i)_{i=1}^n$, along with a permuted weight vector $\theta^{(n)} = \tau(W^{(n)})$ such that $|f^{NN}(x) - f^*(x)| \leq \varepsilon$ for all $x \in [0,1]$.*

Although Theorem 1 can be viewed as a corollary of Theorem 2, the proof process will reveal the practical usefulness of learnable $\alpha, \gamma$ in reducing the required network width to achieve UAP. Moreover, the result can be generalized to the scenario with random initialization, which is stated as

**Theorem 3** (UAP for randomly initialized parameters). *Given a probability threshold $\delta \in (0,1)$, for any function $f^*(x) \in C([0,1])$ and any small number $\varepsilon > 0$, there exists a large even number $n \in \mathbb{Z}^+$, and $\alpha, \gamma \in \mathbb{R}$ for $f_r^{NN}$ in Eq. (2) with randomly initialized $B_r^{(n)} \sim \mathcal{U}[0,1]^n$ and pairwisely randomly initialized $W_r^{(n)} = (\pm p_i)_{i=1}^n$, $p_i \sim \mathcal{U}[0,1]$, along with a permuted weight vector $\theta^{(n)} = \tau(W_r^{(n)})$, such that with probability $1 - \delta$, $|f_r^{NN}(x) - f^*(x)| \leq \varepsilon$ for all $x \in [0,1]$.*

## 2.3 Proof ideas

To identify the UAP of our network (2) in $C([0,1])$, we employ a piecewise constant function, which is a widely-used continuous function approximator (Stein & Shakarchi, 2009), and can be expressed as a summation of several step functions. Next, we demonstrate that our networks can approximate each step function. In this spirit, our constructive proof includes three steps:

1. Approach the target function $f^*$ by a piecewise constant function $g$;
2. Approximate each step function of $g$ by a subnetwork of $f^{NN}$ with permuted coefficients;
3. Annihilate the unused basis functions and coefficients of $f^{NN}$.

Thanks to the Stone-Weierstrass theorem in function approximation theory (Stone, 1948), step 1 can be achieved by dividing the range of $f^*$ with a uniform height to construct each step functions $f_s$ (see Fig. 1(a)). The statement is outlined below (refer to App. B for detailed definition and proof),

**Lemma 1.** *For any function $f^*(x) \in C([0,1])$ and any small number $\varepsilon > 0$, there is a piecewise constant function $g(x)$ with a uniform height $\Delta h \leq \varepsilon$, such that $|g(x) - f^*(x)| \leq \varepsilon$ for all $x \in [0,1]$.*

The execution of step 2 is inspired by the divide-and-conquer algorithm in computer science (Hopcroft et al., 1983) and the multi-grid method in numerical analysis (Hackbusch, 2013). Suppose that the piecewise constant function $g$ in Lemma 1 is a summation of $J$ step functions $\{f_{s_j}\}_{j=1}^J$, we partition the basis functions $B^{(n)}$ also into $J$ subgroups as $B^{(n)} = \cup_{j=1}^J B_j$. Each subgroup $B_j$ contains $b_i$ distributed over the entire domain, instead of localized $b_i$ (see Fig. 1(b)). This allows each subgroup to approach $f_s$ at arbitrary locations using the same construction.

---

[1] Fig. 7 in App. M intuitively shows various kinds of $f^{NN}(x)$ under different permutations.

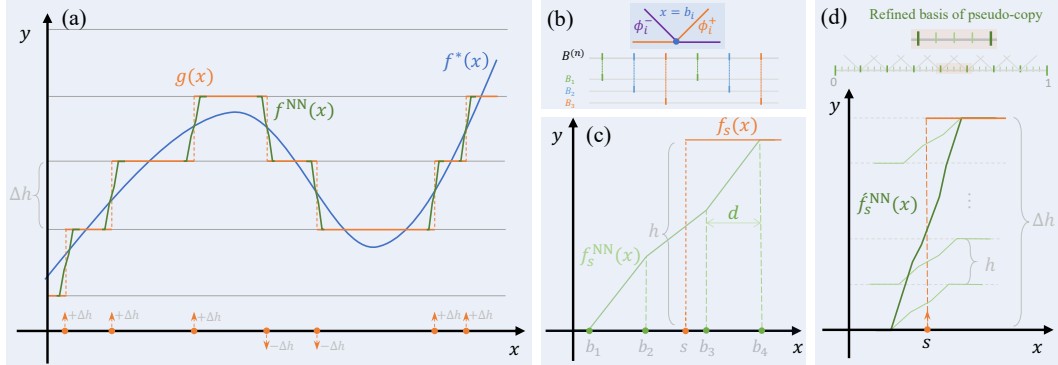

Figure 1: Main idea of the construction. (a) Approximate the continuous function $f^*$ by a piecewise constant function $g$ which is further approximated by permuted networks $f^{\text{NN}}$. (b) Partition of basis functions. (c) The step function approximator $f_s^{\text{NN}}$ constructed by four-pair of basis functions located at $b_1, b_2, b_3, b_4$. (d) Summing pseudo-copies to adjust the heights of resulting function $\acute{f}_s^{\text{NN}}$.

Then, for every subgroup $B_j$, we construct a step function approximator $f_{s_j}^{\text{NN}}$ to approximate $f_{s_j}$, then sum them up to approach $g$. A core technique of this construction is utilizing four pairs of basis functions $\{\pm b_i\}_{i=1}^4$ (shown in Fig. 1(c)), along with a one-to-one correspondence between coefficients and biases (i.e., $\{p_i, q_i\}_{i=1}^4 = \{\pm b_i\}_{i=1}^4$) to meet the permutation training setting, where each coefficient is used only once. This construction can also prevent conflict between different $B_j$.

It is important to note that step 3 is necessary to achieve the desired construction. A crucial challenge of permutation training is that we must assign every parameter, rather than just pick up the wanted parameters and discard the rest. Therefore, it is essential to eliminate the remaining network parameters after step 2 to prevent the potential accumulation of errors. We solve this problem by a processing method that reorganizes them into a linear function with controllable slope and intercept.

To further enhance the conclusion of Theorem 1 to Theorem 2, we introduce a technique called *pseudo-copy*, which can achieve UAP without the linear layer. By refining the parameters distribution, several pseudo-copies $\acute{f}_s^{\text{NN}}$ of the original approximator $f_s^{\text{NN}}$ can be produced with a controllable error (see Fig. 1(d)). The final height can then be adjusted by stacking these copies together, making the scale parameters $\alpha, \gamma$ in Theorem 1 removable.

Extending the UAP to the random initializations is justified by that as the width increases, the parameters randomly sampled from uniform distributions become denser, thus approaching the equidistant case. Therefore, a sufficiently wide network has a high probability of finding a subnetwork that is close enough to the network with UAP in the equidistant case. Then this subnetwork can also achieve UAP due to its continuity. The remaining part of the network can be eliminated by step 3.

## 3 UAP OF PERMUTATION-TRAINED NETWORKS

This section provides a detailed construction of the approximator with weight-permuted networks in the equidistant case, along with an estimation of the convergent rate of approximation error. The extension to the scenario with random initialization is also thoroughly discussed.

### 3.1 THE FOUR-PAIR CONSTRUCTION OF STEP FUNCTION APPROXIMATORS

We start with the equidistant case, and consider four pairs of basis functions $\{\phi_i^{\pm}\}_{i=1}^4$ in Eq. (1) and coefficients $\{p_i, q_i\}_{i=1}^4 = \{\pm b_i\}_{i=1}^4$, where $b_1 \leq b_2 \leq b_3 \leq b_4$ along with a symmetric distance $d = b_2 - b_1 = b_4 - b_3$. The step function approximator $f_s^{\text{NN}}$ has a piecewise linear form as

$$f_s^{\text{NN}}(x) = \sum_{i=1}^4 p_i \phi_i^+(x) + \sum_{i=1}^4 q_i \phi_i^-(x). \tag{3}$$

To ensure a local error of the approximator, we appeal $f_s^{\text{NN}}$ to be $x$-independent outside the interval $[b_1, b_4]$. As a result, the coefficients $p_i, q_i$ must satisfy $\sum_{i=1}^{4} p_i = \sum_{i=1}^{4} q_i = 0$, which implies the correspondence between $\{p_i, q_i\}_{i=1}^{4}$ and $\{\pm b_i\}_{i=1}^{4}$ as

$$\begin{cases} p_1 = -b_1, & p_2 = +b_2, & p_3 = +b_3, & p_4 = -b_4, \\ q_1 = +b_4, & q_2 = -b_3, & q_3 = -b_2, & q_4 = +b_1. \end{cases} \tag{4}$$

and the detailed expression of $f_s^{\text{NN}}$ is given in Eq. (9) at App. C. Notice that $f_s^{\text{NN}}$ is monotone and centrally symmetric about the point $x = \frac{b_2 + b_3}{2}$. So the abstract value of the two constant pieces $x < b_1$ and $b_4 \le x$ are the same. Then the height $h$ of $f_s^{\text{NN}}$ gives

$$h = 2(b_1^2 - b_2^2 - b_3^2 + b_4^2) = 4(b_2 b_3 - b_1 b_4) = 4d(b_4 - b_2). \tag{5}$$

Along with a shifting scale $h/2$, it can approach step function $f_s(x) = h\chi(x - s)$ with $s \in [b_1, b_4]$, where $\chi(z) = 1$ when $z > 0$ and $\chi(z) = 0$ otherwise (see Fig. 1(c)). An example is plotted in Fig. 8 in App. M. It is obvious that the error $\| \left( f_s^{\text{NN}} + h/2 \right) - f_s \|_{L^\infty}$ has a trivial bound $h$.

## 3.2 ANNIHILATE THE UNUSED PART OF THE NETWORK

After constructing step function approximators, the remaining parameters must be suitably arranged to eliminate their impact. Notice that a pair of basis functions $\phi_i^{\pm}$ at each location $b_i$ are either used together or not at all. Therefore, for each unused pair of $\phi_i^{\pm}$ and the corresponding coefficients $\pm p_i$, we can form a linear function $a_i \ell_i$, where $\ell_i(x) := p_i \phi_i^{+}(x) - p_i \phi_i^{-}(x) = p_i x - p_i b_i$ along with a freely adjusted sign $a_i = \pm 1$. The goal then is to choose a proper sign $\mathbf{a} = \{a_i\}_{i=1}^{n}$ for each $\ell_i$ to control $\|\mathcal{S}_\ell\|_{L^\infty}$ in $[0, 1]$, where $\mathcal{S}_\ell(x) := \sum_{i=1}^{n} a_i \ell_i(x)$ is the summed function. It can be achieved by bounding the slope $\sum_{i=1}^{n} a_i p_i$ with respect to $\mathbf{a}$, which becomes a problem of organizing addition and subtraction operations within a given series to reduce the final result. The following lemma provides a solution with an upper bound related to the largest gap in the series.

**Lemma 2.** *For an even number $\bar{n}$ and a sequence of real number $\{c_i\}_{i=1}^{\bar{n}}$ with $c_i \in [0, 1], i = 1, 2, \cdots, \bar{n}$, there exists a combination of sign $\{a_i\}_{i=1}^{\bar{n}}$ with $a_i = \pm 1$, such that $0 \le \sum_{i=1}^{\bar{n}} a_i c_i \le \Delta c$, where $\Delta c = \max_i |c_{i+1} - c_i|$ is the largest gap between the elements in $\{c_i\}_{i=1}^{\bar{n}}$.*

We prove the Lemma 2 by proposing a certain processing method (refer to App. D). As the network width increases, the distribution of $p_i$ will become more dense, causing the largest gap $\Delta p \to 0$, thus the error introduced by the unused part can be arbitrarily small. Notice that the only assumption of this method is the pairwise initialization of coefficients like $(\pm p_i)_{i=1}^{n}$, enabling the extension to random initializations. Besides, it also permits generalization to deeper networks by constructing an identity function and eliminating the remaining parts. Further details can be found in App. D.

## 3.3 APPROXIMATE PIECEWISE CONSTANT FUNCTIONS

Now we briefly discuss how to permute equidistant coefficients $W^{(n)}$ in $f^{\text{NN}}(x) = \sum_{j=1}^{J} f_{s_j}^{\text{NN}}(x)$ to approximate piecewise constant function $g(x) = \sum_{j=1}^{J} a_j \Delta h \chi(x - s_j)$ in Lemma 1 with accuracy $\varepsilon$, where $a_j = \pm 1$ and $\Delta h < \varepsilon/2$. The detailed proof is provided in App. E. We choose $n$ sufficiently large to ensure that every approximator $a_j [f_{s_j}^{\text{NN}}(x) + \frac{h}{2}]$ can approximate $f_{s_j}(x) = a_j h \chi(x - s_j)$ with error $h$. Since the height $h$ in Eq. (5) may not equal $\Delta h$, a multiplying factor $\gamma = \Delta h/h$ is needed. Similarly, the accumulated $h/2$ shifting in each $f_{s_j}^{\text{NN}}$ requires another scaling parameter $\alpha$. Then the whole approximation, along with Lemma 1, allow us to prove the Theorem 1 since

$$\left| f^{\text{NN}}(x) - g(x) \right| = \left| \alpha + \gamma \sum_{i=1}^{n} \left[ p_i \phi_i^{+}(x) + q_i \phi_i^{-}(x) \right] - g(x) \right| \le \Delta h < \varepsilon/2, \quad \forall x \in [0, 1]. \tag{6}$$

Next, we achieve UAP without the scaling parameters $\alpha, \gamma$. The shifting scale $\alpha$ can become small enough by constructing a constant function with a similar height (see App. F). To handle the mismatch between $h$ and $\Delta h$, we introduce the pseudo-copy technique, which stacks $M$ copies of $f_s^{\text{NN}}$ to reach the height $\Delta h = Mh$ (see Fig. 1(d)). However, the copies' locations cannot be identical since the biases $B^{(n)}$ are uniquely assigned. Therefore, we refine the biases $M$-times and partition it into $M$ subgroups as $B^{(Mn)} = \cup_{l=1}^{M} B_l$ like Fig. 1(b). The pseudo-copy $\acute{f}_{s_l}^{\text{NN}}$ is then organized on

each $B_l$, respectively. Since the pseudo-copies are very close to the original one, the refined approximation error $\|\hat{f}_{s_l}^{\mathrm{NN}} - f_s\|_{L^\infty}$ can also be controlled (refer to App. G). Theorem 2 can be proved as below, which indicates that constructing pseudo-copies requires a much larger network.

$$|\hat{f}^{\mathrm{NN}}(x) - g(x)| = \left| \sum_{i'=1}^{Mn} [p_{i'}\phi_{i'}^+(x) + q_{i'}\phi_{i'}^-(x)] - g(x) \right| \leq \Delta h < \varepsilon/2, \quad \forall x \in [0,1]. \tag{7}$$

### 3.4 ESTIMATE THE APPROXIMATION RATE

Here we estimate the approximation rate roughly by the $L^2$ error $E_s$ of approximating single step function $f_s(x) = h\chi(x - s)$ by $f_s^{\mathrm{NN}}(x)$. Start with our four-pair construction in Eq. (3), assume $s = (b_2 + b_3)/2$ and rewrite the relations $b_1 = s - k_2, b_2 = s - k_1, b_3 = s + k_1, b_4 = s + k_2$, where $0 < k_1 \leq k_2$, then the error of single approximator gives (see App. H for details and a similar estimation for pseudo-copies)

$$e_s^2 = \left\| \left( f_s^{\mathrm{NN}} + \frac{h}{2} \right) - f_s \right\|_{L^2}^2 = \frac{8}{3}(k_1 - k_2)^2 \left( k_1^3 + 3k_1^2 k_2 + 2k_1 k_2^2 + k_2^3 \right) \leq \frac{56}{3}d^2 k_2^3. \tag{8}$$

In our step function approximator in Eq. (4), the $k_2$ can be chosen as $k_2 \sim \mathcal{O}(d)$, which implies $e_s \sim \mathcal{O}(d^{5/2})$. However, the height $h$ in Eq. (5) also gives $h \sim \mathcal{O}(d^2)$. To approximate the step function $f_s$ with height $\Delta h \sim \mathcal{O}(1)$, the number of stacked pseudo-copy must satisfy $M = \frac{\Delta h}{h} \sim \mathcal{O}(d^{-2})$. Hence the final error is estimated as $E_s = Me_s \sim \mathcal{O}(d^{1/2})$. Recall that $d = \frac{1}{2n-1}$, we have $E_s \sim \mathcal{O}(n^{-1/2})$, which means the approximation rate is roughly 1/2 order with respect to the network width. We will verify this rate by the experimental results in Sect. 4.

### 3.5 GENERALIZE TO THE RANDOM INITIALIZATIONS

In extending the UAP to the common scenario involving random initializations, the basic proof ideas remain unchanged. However, constructing of step function approximators in Eq. (3) becomes invalid because the desired basis function cannot be located accurately. Nevertheless, the randomly sampled basis functions will become more dense upon increasing width, leading to a high probability of finding basis functions that closely match the required location.

Therefore, we can first apply the UAP in the equidistant case to obtain a network $f^{\mathrm{NN}}$ in Eq. (2), which exhibits approximation power. Then, within a randomly initialized network of sufficient width, we find a subnetwork $f_{\mathrm{sub}}^{\mathrm{NN}}$ that can be regarded as randomly perturbed from $f^{\mathrm{NN}}$. If this perturbation is small enough, the subnetwork $f_{\mathrm{sub}}^{\mathrm{NN}}$ will also possess approximation power.

Notice that this declaration can hold for totally random coefficients $W_r^{(n)} \sim \mathcal{U}[0,1]^{2n}$. However, eliminating the unused parameters by the process discussed in Sect. 3.2 requires a pairwise form such as $W_r^{(n)} = (\pm p_i)_{i=1}^n$. Therefore, we restrict our result to the case in Theorem 3. The detailed proof along with an estimation of the probability introduced by randomness are given in App. I.

## 4 EXPERIMENTS

This section presents numerical evidence to support and validate the theoretical proof. An interesting observation of permutation behaviors also highlights the theoretical potential of this method.

### 4.1 THE ALGORITHM IMPLEMENTATION OF PERMUTATION TRAINING

In the implementation of permutation training, guidance is crucial in finding the ideal order relationship of the weights. The *lookahead permutation (LaPerm)* algorithm proposed in Qiu & Suda (2020) introduces an $k$-times Adam-based free updating (Kingma & Ba, 2015), where the learned relationship can then serve as a reference for permuting. To ensure the performance, the weights are permuted after every $k$ epoch. Apart from the fixed permutation period $k$ chosen by Qiu & Suda (2020), we also consider a relaxed algorithm with a gradually increased $k$ to learn sufficient information for the next permutation. The impact of $k$'s value on convergence behavior is evaluated to be negligible (see App. N). See App. J for a discussion of the original and relaxed LaPerm algorithms.

## 4.2 EXPERIMENTAL SETTING OF FUNCTION APPROXIMATION TASKS

Now we carry out experiments for some regression problems to justify our theoretical results. We consider a three-layer network in Eq. (2), where the first hidden layer's parameters are fixed to form the ReLU basis functions $\{\phi_i^{\pm}\}_{i=1}^n$ in Eq. (1), and the weights $\theta^{(n)}$ of the second hidden layer are trained by permutation. Moreover, $\alpha, \gamma$ in the output layer are freely trained scaling factors to reduce the required network width. All the experiments below are repeated 10 times with different random seeds, and the error bars mark the range of the maximum and minimum values. Refer to App. L for the detailed experimental environment and setting for each case.

## 4.3 APPROXIMATING THE ONE-DIMENSIONAL CONTINUOUS FUNCTIONS

For one-dimensional cases, we utilize a 1-2$n$-1-1 network architecture with random initializations discussed in Theorem 3. The approximation targets are typical continuous functions $y = -\sin(2\pi x)$, and 3-order Legendre polynomial $y = \frac{1}{2}(5x^3 - 3x)$, where $x \in [-1, 1]$. A more complicated case about a Fourier series with random coefficients, along with the results of the equidistant scenario, are presented in App. Q. The numerical result illustrated in Fig. 2 exhibits a clear convergence behavior upon increasing $n$. Our relaxed LaPerm algorithm doesn't show a significant advantage, potentially due to the preliminary attempt of exponentially increasing $k$. This suggests a need for advanced relaxation schemes, such as a self-adjusted strategy (Qiao et al., 2011). Furthermore, the $L^\infty$ error exhibits a 1/2 convergence rate with respect to $n$. Although the theoretical estimation in Sect. 3 is based on $L^2$ norm, we indeed observe that it also holds for $L^\infty$ error.

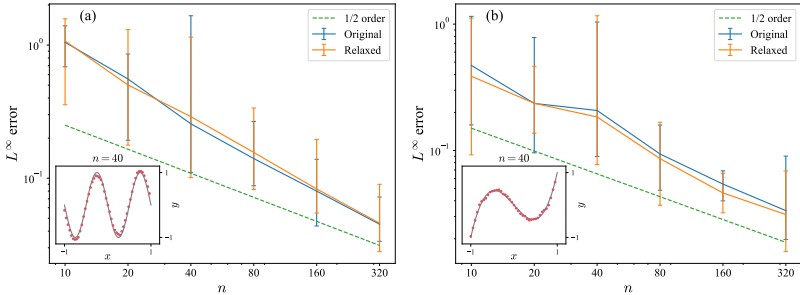

Figure 2: Approximating one-dimensional continuous function (a): $y = -\sin(2\pi x)$ and (b): $y = \frac{1}{2}(5x^3 - 3x)$ with randomly initialized network, where $x \in [-1, 1]$. The inset in each panel presents the target function as lines and an example of the approximation result as dots.

## 4.4 THE PERFORMANCE OF VARIOUS RANDOM INITIALIZATIONS

Here we discuss the impact of initialization on performance, which is more crucial for permutation training due to the weights' lack of positional flexibility. Here we utilize the case in Fig. 2(a) to consider 8 different random initialization methods. Fig. 3 shows that the UAP in permutation-trained networks is not limited in the setting considered by our theorems. The converged random cases followed the pairwise initialization outperform the equidistant scenario, demonstrating the well-known advantages of random initializations. However, some commonly used random initializations, such as Xavier's uniform initialization $U_X$ (Glorot & Bengio, 2010), and He's uniform and normal initializations $U_H$ and $N_H$ (He et al., 2015), fail to show convergence behavior. These results emphasize the incompatibility between the existing initializations and the permutation training setting.

Further insight can be found by comparing the results in pairs. We first focus on totally and pairwisely randomly initializing $W^{(n)}$ from uniform distribution $\mathcal{U}[-1, 1]$, which are labeled as case 1 and 2, respectively. Apart from the clear dominance of pairwise case 1, the total case 2 also shows a certain degree of approximation power. Next, for a randomly initialized $B^{(n)}$, in case 3 we let $W^{(n)}$ have a strict correspondence like the equidistant case, while in case 4 $W^{(n)}$ is initialized separately. The almost equivalent results indicate that the correspondence between $B^{(n)}$ and $W^{(n)}$ in Eq. (3) may not be necessary in random cases. Moreover, we apply the standard $U_H$ for $W^{(n)}$ in case 5

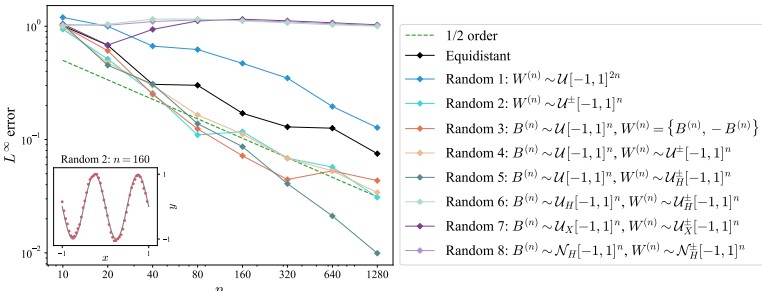

Figure 3: The performance of randomly initialized parameters to approximate $y = -\sin(2\pi x)$, where $x \in [-1, 1]$. The pairwise random distribution of $W^{(n)} = (\pm p_i)_{i=1}^n$, $p_i \sim \mathcal{U}[-1, 1]$ is noted as $W^{(n)} \sim \mathcal{U}^{\pm}[0, 1]^n$, and the same applies to $\mathcal{U}_X^{\pm}[0, 1]^n$ and $\mathcal{N}_H^{\pm}[0, 1]^n$. The error bars are omitted for conciseness. The inset panel presents the target function as lines and an example of the approximation result as dots.

and also for $B^{(n)}$ in case 6. It shows that case 5 achieves the best accuracy for larger networks ($n > 320$), while case 6 exhibits unexpected deterioration, which may be attributed to the mismatch of the scale in $B^{(n)}$. Finally, the default choices $N_H$ and $U_X$ in cases 7 and 8 both yield surprisingly poor performance, underscoring the need for new initializations suitable to permutation training.

## 4.5 Observation of the Permutation-Active Patterns

This section aims to explore the theoretical potential of permutation training in describing network learning behavior. Based on the significant correlation between permutation and learning behavior as evidenced by Qiu & Suda (2020) and our investigation, we hypothesize that the permutation-active components of the weight vector play a crucial role in the training process. Therefore, by identifying and tracing the permutation-active part of weights, a novel tool that provides insights into learning behavior can be achieved, which also facilitates visualization and statistical analysis.

As a preliminary attempt, we illustrate the permutation behavior of the coefficients $\theta^{(n)}$ in Fig. 4. The components that participated in the permutation are visually highlighted in dark green. The behavior clearly demonstrated that the order relationship evolves synchronously with the learning process, agreeing with the observation in Qiu & Suda (2020).

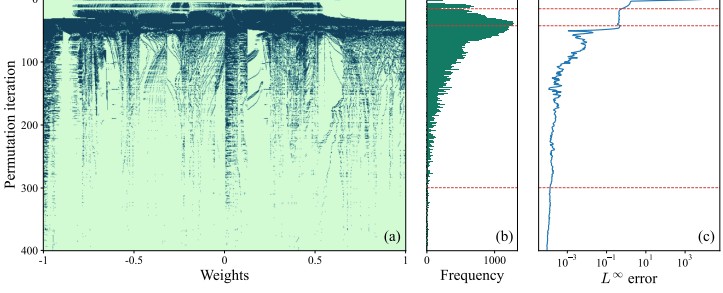

Figure 4: The permutation behavior in the first 400 permutation iteration in approximating $y = -\sin(2\pi x)$ by equidistantly initialized network with $n = 640$. (a) The distribution of the active components (denoted by dark green color). (b) The frequency distribution illustrates the variation in the total count of active components in each permutation. (c) The corresponding loss behavior.

Specifically, the distribution of the active components shows significant patterns, which can be classified into four stages (marked in red dash lines in Fig. 4). The loss declines sharply in the initial stage, while only the components with medium value are permuted. Once loss reaches a plateau in the second stage, more components are involved in permutation, evidencing the role of permutation

in propelling the training. As loss starts to decline again, the permutation frequency correspondingly diminishes. Interestingly, the slower loss decrease gives rise to a ribbon-like pattern, akin to the localized permutations reported by Qiu & Suda (2020). This is possibly due to slow updates failing to trigger a permutation. This observation may support the existence of inherent low-dimensional structures within the permutation training dynamics, potentially linked to mathematical depiction of permutation groups, such as cycle decomposition (Cameron, 1999) and Fourier bases for permutation (Huang et al., 2009). Finally, the permutation's saturation stage aligns with the stationary state of loss convergence. We believe these inspiring phenomena deserve further exploration.

## 5 CONCLUSION AND DISCUSSION

As a constrained training method, permutation training exhibits unique properties and practical potential (see App. A). To verify its efficacy, we prove the UAP of permutation-trained networks with equidistant initialization and pairwise random initialization for one-dimensional continuous functions. The key idea is a four-pair construction of step function approximators in Fig. 1, along with a processing method to eliminate the impact of the remaining parameters. Our experimental results not only confirm the theoretical declarations (see Fig. 2), but also validate the approximation power for various random initializations in Fig. 3, establishing the prevalence of the UAP of permutation training. The discovery that certain commonly used initializations fail to achieve UAP also raises an intriguing question about the systematical characterization of initializations that satisfy UAP.

The generalizability of our results holds significant importance. Extending to networks equipped with leaky-ReLU can be straightforward (refer to App. O for numerical evidence). Our approach also facilitates implementations within other architectures (see App. P for detailed discussion). However, extending our results to the high-dimensional scenario still faces some theoretical challenges, although some preliminary experimental attempts have been made for two-dimensional inputs (see App. K). One potential approach is similar to the discussion in Sect. 3.5, but here we can directly seek the subnetwork as a random perturbation from the network with conventional UAP in high dimensions. To achieve this, however, the processing method in Sect. 3.2 must be generalized from pairwise to total random initializations. We plan to address this problem in future work.

Our observation in Sec. 4.5 suggests that permutation training is a novel tool to shed light on network learning behavior. It corresponds well with the training process and has systematical mathematical descriptions (Cameron, 1999; Huang et al., 2009). Specifically, the patterns observed in Fig. 4 can intuitively justify some weight categorization strategies, leading to potential benefits for consolidating the crucial weights for previous tasks (Maltoni & Lomonaco, 2019), or pruning to find the ideal subnetwork in the lottery ticket hypothesis (Frankle & Carbin, 2019). Additionally, the existing permutation training algorithm can be viewed as applying an order-preserving projection from the free training results to the initial weight value, sharing the same form as weight projection methods in continual learning (Zeng et al., 2019).

This work is expected to facilitate the practical applications of permutation training. However, some issues still exist and deserve further investigation. Notably, existing initializations derived from the free training situation, such as He's normal initialization, perform poorly with permutation training in Fig. 3, emphasizing the need for developing more compatible initializations. This could pave the way to effectively training higher-dimensional and deeper networks by weight permutation, thereby meeting the practical requirements. Further, the permutation training itself also has the potential to serve as an initialization protocol (Scabini et al., 2022).

The existing attempts at algorithm implementations guide the permutation by Adam-based inner loops, thus incurring undesirable external computation costs. However, if the order relationships can be learned through other time-saving approaches, such as the learn-to-rank formalism (Cao et al., 2007), or permutation search algorithms in the study of LMC (Jordan et al., 2023; Ainsworth et al., 2023), the benefits of permutation training will be actualized in practice. Importantly, our proof is independent of algorithm implementations, which is expected to inspire and motivate the development of more advanced algorithms.

Overall, we believe that the UAP of permutation-trained networks underscores the profound, yet undiscovered insights into how the weight encodes the learned information, highlighting the importance of further exploration in this field.

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

# APPENDIX

## A PERMUTATION TRAINING'S ADVANTAGES IN HARDWARE IMPLEMENTATION

This section gives a detailed discussion about the practical potential of permutation training on specific hardware implementations, which also highlights the limitation of the current CPU or GPU-based hardware in fully unleashing its hardware-friendly properties.

Although permutation training can achieve comparable or even slightly better performance for image classification tasks (Qiu & Suda, 2020), this method may not take a significant advantage in terms of purely performance comparison on current CPU or GPU-based hardware, compared with the conventional free-training methods such as Adam (Kingma & Ba, 2015). This is mainly due to the strong constraints posed by the extreme setting of permutation training. Besides, these general-purpose architectures have not been specifically fine-tuned for this method, which is further compounded by the lack of appropriate algorithms and initialization as discussed in Sect. 5.

However, we believe it is highly suitable for the design of physical neural networks (Nugent, 2005). The benefits stem from the fact that in the realm of physical neural networks, reconnecting the neurons is sometimes more convenient than changing the exact value. Therefore, permutation training may inspire alternative physical weight connection implementations, such as using fixed-weight devices controlled by a permutation circuit (Qiu & Suda, 2020).

One promising application focuses on reconfigurability, since building large physical neural networks while ensuring reconfigurability is challenging (Nugent, 2005). The permutation circuit enables the deployment and production of reconfigurable physical neural networks at a lower cost. Such an approach has a close connection with the realm of continual learning (Maltoni & Lomonaco, 2019), and has already been applied to a fixed-weight network accelerator (Kosuge et al., 2021a;b).

Another kind of potential application scenario is the physical neural networks with an explicit structure to store the weight value, such as the *integrated photonic tensor core*, a computing chip with specialized architecture (Feldmann et al., 2021). This design has been successfully employed by international commercial companies in their photonic computing products. In each photonic tensor core, an array of phase-change cells are organized to separately store each element of the weight matrix, with their values adjusted through optical modulation of the transmission states of the cells. However, permutation training indicates that, in addition to changing the exact value, it is feasible to connect each cell with the permutation circuit for convenient reconnections. Therefore permutation training can facilitate the learning process.

## B PROOF OF LEMMA 1

The complete statement of Lemma 1 is as follows.

**Lemma 1.** *For any function $f^*(x) \in C([0,1])$ and any small number $\varepsilon > 0$, there is a piecewise constant function $g(x)$ with a common jump $\Delta h \leq \varepsilon$,*

$$g(x) = \Delta h \sum_{j=1}^{J} \alpha_j \chi(x - s_j), \quad \alpha_j \in \{-1, +1\}, \quad J \in \mathbb{Z}^+,$$

*such that $|g(x) - f^*(x)| \leq \varepsilon$ for all $x \in [0,1]$. Here $\chi(x)$ is the standard step function*

$$\chi(x) = \begin{cases} 0, & x < 0, \\ 1, & x \geq 0. \end{cases}$$

*Proof.* The function $g(x)$ can be constructed explicitly. According to the Stone-Weierstrass theorem (Stone, 1948), we can assume $f^*$ to be a polynomial function for simplicity. Let the range of $f^*$ be covered by an interval $[\beta_1 \Delta h, \beta_2 \Delta h]$ with two integers $\beta_1, \beta_2 \in \mathbb{Z}$. Then take $s_j$ as the values such that $f^*(s_j) = (k + 0.5)\Delta h$ for some $k \in \mathbb{Z}$. The sign of $\alpha_j$ is determined according to the values of $f^*(x)$ on $[s_{j-1}, s_{j+1}]$. It is easy to verify such construction satisfies our requirements. □

## C  EXPLICIT EXPRESSION OF STEP FUNCTION APPROXIMATOR $f_s^{\mathrm{NN}}$

By substituting the choice of coefficients $\{p_i, q_i\}_{i=1}^4$ in Eq. (4) into Eq. (3) of $f_s^{\mathrm{NN}}$ , the piecewise form gives

$$f_s^{\mathrm{NN}}(x) = \begin{cases} 2b_1 b_4 - 2b_2 b_3 & -1 \le x < b_1, \\ (-b_1 + b_4)x + b_1^2 + b_1 b_4 - 2b_2 b_3 & b_1 \le x < b_2, \\ (-2b_1 + 2b_2)x + b_1^2 - b_2^2 + b_1 b_4 - b_2 b_3 & b_2 \le x < b_3, \\ (-b_1 + b_4)x + b_1^2 - b_2^2 - b_3^2 + b_1 b_4 & b_3 \le x < b_4, \\ b_1^2 - b_2^2 - b_3^2 + b_4^2 & b_4 \le x \le 1. \end{cases} \tag{9}$$

Although it was suggested by Cai et al. (2021) that a step function could be expressed as a combination of two ReLU basis functions, this approach is unsuitable for the approximation of the piecewise constant function $g$ in Lemma 1. The reason is as below.

Note a step function at $x = s$ with height $h_s$ as $f_s(x) = h_s \chi(x - s)$. Then $f_s$ can be well approximated by $\tilde{f}_s$ , which is a linear combination of two ReLU bases,

$$\tilde{f}_s(x) = \frac{h_s}{2\epsilon}[\mathrm{ReLU}(x - b_i) - \mathrm{ReLU}(x - b_j)],$$

where $\epsilon$ is the error tolerance, and the biases are $b_i = s - \epsilon$ and $b_j = s + \epsilon$, respectively. Although the construction of $\tilde{f}_s$ is similar to our network (2), it is not entirely suitable for us because the coefficients are fully determined by the height of $f_s$. Since our coefficients differ initially, this method can not approach several step functions of the same height, which hinders approximating the continuous functions with corresponding step functions.

## D  ELIMINATE THE IMPACT OF UNUSED PARAMETERS IN NETWORKS

Here we aim to eliminate the impact of the unused part in the network. Firstly we consider the equidistant case with $W^{(n)} = (\pm b_i)_{i=1}^n$. Suppose that there is a total $\bar{n}$ pair of unused basis functions, where $\bar{n}$ is an even number. Notice that at each location $b_i$, we have a pair of basis functions $\phi_i^{\pm}$, which are either used together or not at all. Therefore, for each unused pair of basis functions $\phi_i^{\pm}$ and the corresponding weights $\pm b_i$, we can form a linear function $a_i \ell_i(x)$ where

$$\ell_i(x) := b_i \phi_i^+(x) - b_i \phi_i^-(x) = b_i x - b_i^2, \tag{10}$$

and the sign $a_i = \pm 1$ can be freely adjusted. The goal is to choose a proper sign $\mathbf{a} = \{a_i\}_{i=1}^{\bar{n}}$ for each linear function $\ell_i$ to control $\max_{x \in [0,1]} |\mathcal{S}_\ell(x)|$, the $L^\infty$-norm of the resulting function after taking a sum, where

$$\mathcal{S}_\ell(x) = \sum_{i=1}^{\bar{n}} a_i \ell_i(x) = \sum_{i=1}^{\bar{n}} a_i b_i x - \sum_{i=1}^{\bar{n}} a_i b_i^2. \tag{11}$$

This problem is equivalent to controlling the slope of the resulting function $\sum_{i=1}^{\bar{n}} a_i b_i$ with respect to $\mathbf{a}$ when $b_i \in [0, 1]$, since it will simultaneously lead to an upper bound of the intercept $\sum_{i=1}^{\bar{n}} a_i b_i^2$. Therefore, the problem transforms into how to organize addition and subtraction operations of a given series of numbers to reduce the abstract value of the final result. Lemma 2 provides a solution with an upper bound related to the distance between the basis functions.

*Proof of Lemma 2.* The basic idea is applying the *Leibniz's test* (known as *alternating series test*) (Rudin, 1953) in mathematical analysis, which guarantees that an alternating series that decreases in absolute value is bounded by the leading term. However, utilizing Leibniz's test directly to $\{c_i\}_{i=1}^{\bar{n}}$ can only get a trivial upper bound $\sum_{i=1}^{\bar{n}} a_i b_i \le c_1 \le 1$. Therefore, we need to first subtract them pairwise to obtain a new sequence, where each element represents the gap between the elements in $\{c_i\}_{i=1}^{\bar{n}}$, and then apply the Leibniz's test to get the desired bound.

Firstly we assume $1 \geq c_1 \geq c_2 \geq \cdots \geq c_{\bar{n}} \geq 0$, and note $\bar{n} = 2\tilde{n}$ since it is an even number, which is guaranteed from that $\bar{n}$ is an even number, and the number of basis functions used in our construction is always a multiple of 4. Then we can define a new series $\{r_i\}_{i=1}^{\tilde{n}}$ as

$$r_i = c_{2i-1} - c_{2i}, \quad i = 1, 2, \cdots, \tilde{n}.$$

Then we rearrange $\{r_i\}_{i=1}^{\tilde{n}}$ in descending order to have $\{r_i'\}_{i=1}^{\tilde{n}}$, and alternatively use addition and subtraction operations on each element, leading to the result as

$$\mathcal{S}_{\tilde{n}} = r_1' - r_2' + r_3' - r_4' + \cdots \leq r_1' \leq \Delta c$$

Since $c_i \in [0,1], i = 1, 2, \cdots, \bar{n}$, the series $\{r_i'\}_{i=1}^{\tilde{n}}$ is bounded by 1, such that $r_i' \to 0$ as $i \to \infty$. Therefore, according to Leibniz's rule, we know that $\mathcal{S}_{\tilde{n}}$ can be bounded by $r_1' \leq \Delta c$.

Notice that $r_1'$ may not necessarily equal $\Delta c$ since the biggest gap can possibly occur at the even-numbered intervals. An example is for $\{c_i\}_{i=1}^4 = \{100, 99, 2, 1\}$, the corresponding $\{r_i\}_{i=1}^2 = \{1, 1\}$, thus $\mathcal{S}_2 < r_1' = 1$ while $\Delta c = 97$.

Next, for a sequence $\{c_i\}_{i=1}^n$ without certain order and signs, we can rearrange them into a new sequence $\{c_i'\}_{i=1}^n$ in descending order of their value, and then perform the same operation as before.

$\square$

Thanks to Lemma 2, we can construct a linear function with a controllable slope and intercept by the unused part of the network. To estimate the error corresponding to the summation function $\mathcal{S}_\ell$ in Eq. (11), we rewriting $\mathcal{S}_\ell = \beta x + \eta$, and note the $L^\infty$ error by

$$E_u(\beta, \eta) = \max_{x \in [0,1]} |\mathcal{S}_\ell(x)| \leq |\beta| + |\eta|, \quad \beta \in [0, \Delta b], \quad \eta \in [-\Delta b^2, 0]. \tag{12}$$

Then it has the following upper bound as

$$E_u \leq \Delta b(1 + \Delta b). \tag{13}$$

As the network width increases, the basis function will become more dense, leading to $\Delta b \to 0$ thus $E_u \to 0$.

For the case with randomly initialized weights $W_r^{(n)} = (\pm p_i)_{i=1}^n$, the basis functions $\phi_i^\pm$ located at $b_i$ are equipped with weights $\pm p_i$ to ensure that $|b_i - p_i| =: \delta_i$ to be small enough. A possible method is to arrange both $\{b_i\}_{i=1}^{\bar{n}}$ and $\{p_i\}_{i=1}^{\bar{n}}$ in descending order and then pair the elements at corresponding positions. Notice that $p_i$ and $b_i$ utilized by the subnetwork $f_{\text{sub}}^{\text{NN}}$ are also very close to each other. Consequently, for a sufficiently large $\bar{n}$, it is a high probability for each $\delta_i$ to be small enough. because for any small number $\bar{\varepsilon}$, the probability of $\delta_i \leq \bar{\varepsilon}, i = 1, 2, \cdots, \bar{n}$ is $[1 - (1 - 2\bar{\varepsilon})^{\bar{n}}]^{\bar{n}} \leq 1 - (1 - 2\bar{\varepsilon})^{\bar{n}^2}$.

Therefore, the resulting linear function has a similar form to Eq. (10) as $a_i \ell_i^r$, where

$$\ell_i^r(x) := p_i \phi_i^+(x) - p_i \phi_i^-(x) = p_i x - p_i b_i,$$

and the corresponding problem in Eq. (11) gives $\|\mathcal{S}_\ell^r\|_{L^\infty}$, where

$$\begin{aligned}
\mathcal{S}_\ell^r(x) = \sum_{i=1}^{\bar{n}} a_i \ell_i^r(x) &= \sum_{i=1}^{\bar{n}} a_i p_i x - \sum_{i=1}^{\bar{n}} a_i p_i b_i \\
&= \sum_{i=1}^{\bar{n}} a_i p_i x - \sum_{i=1}^{\bar{n}} a_i p_i^2 + \sum_{i=1}^{\bar{n}} a_i p_i \delta_i \\
&=: \beta_r x + \eta_r
\end{aligned}$$

Thus by applying Lemma 2 to the series $\{p_i\}_{i=1}^{\bar{n}}$, we can get a choice of $a_i^0, i = 1, 2, \cdots, n$, which can bound the slope $\beta_r$ as

$$0 \leq \beta_r = \sum_{i=1}^{\bar{n}} a_i^0 p_i \leq \Delta p \tag{14}$$

where $\Delta p$ is the largest gap between the elements in $\{p_i\}_{i=1}^{\bar{n}}$. For the intercept $\eta_r$, we apply summation by parts as

$$
\begin{aligned}
|\eta_r| &\leq \left| -\sum_{i=1}^{\bar{n}} a_i^0 p_i^2 \right| + \left| \sum_{i=1}^{\bar{n}} a_i^0 p_i \delta_i \right| \\
&\leq \Delta p^2 + \left| \delta_{\bar{n}} A_{\bar{n}} - \sum_{j=1}^{\bar{n}-1} A_j (\delta_{j+1} - \delta_j) \right|, \\
&\leq \Delta p^2 + \bar{\varepsilon}\Delta p + \sum_{j=1}^{\bar{n}-1} |A_j| |\delta_{j+1} - \delta_j|.
\end{aligned}
\tag{15}
$$

where $A_j = \sum_{i=1}^{j} a_i^0 (b_i + r_i)$, and $A_{\bar{n}} = \beta_r \leq \Delta p$.

Here, we utilize *Dirichlet's test* (Rudin, 1953) to control the last term on the right-hand side of Eq. (15), which firstly requires an upper bound of $A_j, j = 1, 2, \cdots, \bar{n} - 1$. According to our processing method in Lemma 2, it satisfies

$$
|A_j| \leq \Delta p + 1, \quad j = 1, 2, \cdots, \bar{n} - 1.
$$

since when $j$ is even, $|A_j| \leq |A_{\bar{n}}| \leq \Delta p$. And for odd $j$, $|A_j| \leq |A_{j-1}| + |a_j p_j| \leq \Delta p + \max_i |p_i| = \Delta p + 1$. Next, since only a finite number of terms are being summed in Eq. (15), we can always rearrange the summation to ensure that the rearranged series $\{\delta_{j'}\}_{j'=1}^{\bar{n}}$ is monotonically decreasing, Therefore, by Dirichlet's test, we have

$$
\begin{aligned}
|\eta_r| &\leq \Delta p^2 + \bar{\varepsilon}\Delta p + \sum_{j'=1}^{\bar{n}-1} |A_{j'}| |\delta_{j'+1} - \delta_{j'}| \\
&\leq \Delta p^2 + \bar{\varepsilon}\Delta p + (\Delta p + 1) \sum_{j'=1}^{\bar{n}-1} (\delta_{j'+1} - \delta_{j'}) \\
&\leq \Delta p^2 + \bar{\varepsilon}\Delta p + (\Delta p + 1) 2\bar{\varepsilon} = \Delta p^2 + \Delta p(1 + 2\bar{\varepsilon}) + 2\bar{\varepsilon} =: \Delta\eta_r
\end{aligned}
\tag{16}
$$

Along with the estimation in Eq. (14) and (16), we can write the $L^\infty$ error similarly with Eq. (12) as

$$
E_{u,r}(\beta_r, \eta_r) = \max_{x \in [0,1]} |\mathcal{S}_\ell^r(x)| \leq |\beta_r| + |\eta_r|, \quad \beta_r \in [0, \Delta p], \quad \eta_r \in [-\Delta\eta_r, \Delta\eta_r].
$$

Then it has the following upper bound as

$$
E_{u,r} \leq \Delta p + \Delta p^2 + \Delta p(1 + 2\bar{\varepsilon}) + 2\bar{\varepsilon} = \mathcal{I} + \mathcal{J},
\tag{17}
$$

where

$$
\begin{aligned}
\mathcal{I} &= 2\bar{\varepsilon}, \\
\mathcal{J} &= \Delta p \big[ \Delta p + (2 + 2\bar{\varepsilon}) \big].
\end{aligned}
$$

Notice that $\mathcal{I}$ depends solely on $\bar{\varepsilon}$, while $\mathcal{J}$ has a common factor $\Delta p$. Therefore, we can choose $\bar{\varepsilon}$ to ensure that $\mathcal{I}$ is small enough, then as the network width increases, the basis function will become more dense, leading to $\Delta p \to 0$ thus $e_{u,r}^2$ can be arbitrarily small.

Additionally, this linear function construction also enables extending the UAP to deeper networks, which is not very obvious when using permutation training. Notice that we can construct an identity function $y = x$ by a pair of basis functions $\phi_i^\pm$ as $y = p_i \phi_i^+(x) - p_i \phi_i^-(x)$, where $b_i = 0$, $p_i = 1$. It enables us to approach identity functions with subsequent layers, thus the situation of deep networks is equivalent to the cases discussed in this paper. As a result, we can achieve UAP in deep networks.

## E    DETAILED PROOF OF THEOREM 1

For the piecewise constant function $g = \sum_{j=1}^{J} f_{s_j}$ discussed in Lemma 1, we can denote the $j$-th step function in $g$ as $f_{s_j}(x) = a_j h \chi(x - s_j)$ with $s_j \in [b_{k_j}, b_{k_j+1})$, $a_j = \pm 1$, where $j = 1, \cdots, J$. Then we consider the smallest distance among different step functions in $g$ as a crucial parameter

$$
\delta_0 = \min_{j=1,\cdots,J-1} |s_j - s_{j-1}|.
$$

Therefore, we can choose $n$ sufficiently large to ensure the distance between basis functions $\delta = \frac{1}{2n-1} < \frac{\delta_0}{8}$ to avoid the conflict between step function approximators.

For each $f_{s_j} = a_j h \chi(x - s_j)$, the approximator $f_{s_j}^{\text{NN}}$ in (3) are constructed. We choose the basis functions $\{\phi_k^{\pm}\}_{k=1}^4$ corresponding to $k \in \{k_j - 1, k_j, k_j + 1, k_j + 2\}$. Under the configuration principle in (4), we have $a_j [f_{s_j}^{\text{NN}}(x) + \frac{h}{2}]$ to approximate $f_{s_j}(x) = a_j h \chi(x - s_j)$ with error $h = 8d^2 = 8\delta^2$. Note that $f_{s_j}^{\text{NN}}(x) + \frac{h}{2} = f_{s_j}(x)$ if $x \notin [b_{k_j-1}, b_{k_j+2}]$, where the intervals do not intersect since $\delta < \delta_0/8$ is small enough. As a result, we have

$$\left| \sum_{j=1}^J a_j \left[ f_{s_j}^{\text{NN}}(x) + \frac{h}{2} \right] - \sum_{j=1}^J a_j h \chi(x - s_j) \right| \le h, \quad \forall x \in [0, 1]. \tag{18}$$

Denote $K = \{1, 2, ..., n\}$ as the index of the location $B^{(n)}$, and $K_{\text{used}} = \cup_{j=1}^J \{k_j - 1, k_j, k_j + 1, k_j + 2\}$ for the basis functions used by approximators. Then the Eq. (18) can be rewritten as

$$\left| \sum_{k \in K_{\text{used}}} \left[ p_k \phi_k^+(x) + q_k \phi_k^-(x) \right] + \frac{h}{2} J' - \frac{h}{\Delta h} g(x) \right| \le h,$$

where $J' = \sum_{j=1}^J a_j < J$. Besides, a large enough $n$ can also ensure that the error introduced by the unused parameters

$$E_u = \max_{x \in [0,1]} \left| \sum_{k \in K/K_{\text{used}}} \left[ p_k \phi_k^+(x) + q_k \phi_k^-(x) \right] \right| \le h.$$

Therefore, the whole approximation reads

$$\left| f^{\text{NN}}(x) - g(x) \right|$$
$$\le \left| \frac{\Delta h}{2h} \sum_{k \in K_{\text{used}}} \left[ p_k \phi_k^+(x) + q_k \phi_k^-(x) \right] + \frac{\Delta h}{2} \left( \frac{J'}{2} - \frac{C}{h} \right) - g(x) \right| + \frac{\Delta h}{2h} E_u \le \Delta h, \quad \forall x \in [0, 1]. \tag{19}$$

where we set $\Delta h < \varepsilon, \alpha = \frac{\Delta h}{2}(\frac{J'}{2} - \frac{C}{h}), \gamma = \frac{\Delta h}{2h}$. This allows us to prove the Theorem 1.

*Proof of Theorem 1.* According to Lemma 1, there is a piecewise constant function $g(x)$ with a constant height $\Delta h \le \varepsilon/2$ such that $|g(x) - f^*(x)| \le \varepsilon/2$. Using the construction (19), we have $|g(x) - f^{\text{NN}}(x)| \le \varepsilon/2$. Then we have $|f^{\text{NN}}(x) - f^*(x)| \le \varepsilon$ for all $x \in [0, 1]$. □

## F   THE CONSTRUCTION OF CONSTANT FUNCTION APPROXIMATORS

In the equidistant case, we can construct a step function approximator to provide the necessary shift for each $f_s^{\text{NN}} = C$. It also offers another method to eliminate the impact of the unused part of the network.

Following the same form in Eq. (3), a constant function approximator $f_c^{\text{NN}} = C$ for some constant $C$ can be constructed. The coefficients $\{p_i, q_i\}_{i=1}^4$ are set to equalize the height of the two constant pieces $x < b_1$ and $b_4 \le x$, leading to $-\sum_{i=1}^4 p_i b_i = \sum_{i=1}^4 q_i b_i$. A choice that satisfied these relationships is

$$\begin{cases} p_1 = -b_1, & p_2 = +b_2, & p_3 = +b_3, & p_4 = -b_4, \\ q_1 = +b_1, & q_2 = -b_2, & q_3 = -b_3, & q_4 = +b_4. \end{cases} \tag{20}$$

It gives a representation of constant $C = 2d(b_4 - b_2)$. The symmetry of coefficients vanishes the approximation error, i.e., $\|f_c^{\text{NN}} - f_c\|_\infty = 0$. We can then create a negative constant $-C$ by changing the sign of $p_i, q_i$ in Eq. (20). Since there are $2n$ basis functions in total, we can choose $\{\pm b_i\}_{i=1}^4$ to form a step function approximator, then select another odd or even number of sets for the shifting scale, leaving the unused $b_i$ in pairs to offset with each other.

## G    Detailed proof of Theorem 2

Now we move to the case of fixed $\gamma = 1, \alpha = 0$ by refining the previous construction. Note that the construction in the proof of Theorem 1 needs $\alpha = \frac{\Delta h}{2}(\frac{J'}{2} - \frac{C}{h})$ and $\gamma = \frac{\Delta h}{2h}$, where $h = 8\delta^2 < \delta_0$ can be choosed much smaller than $\Delta h$, leading to a large scaling factor $\gamma \gg 1$. Since $\Delta h$ and $\delta = \frac{1}{2n-1}$ have some flexibility, we can adjust $\gamma$ to be an integer $M \in \mathbb{Z}^+$, $i.e.$ $\Delta h = 2Mh$. It implies that we can stack $f_{s_j}^{\mathrm{NN}}(x)$ in Eq. (3) $M$ times (instead of multiplying by $\gamma$) to match the desired height. Notice that each basis function is used uniquely, here we create some $pseudo$-$copies$ of $f_{s_j}^{\mathrm{NN}}(x)$, which are not entirely equal but very close. The approach is constructed as follows.

Following the previous discussion we construct each $f_{s_j}^{\mathrm{NN}}$ with $\Delta h = 2Mh$, $h = 8\delta^2$ and $\delta = \frac{1}{2n-1}$. Then we control $M = 2m + 1$ as an odd number and refine the basis functions from $f_{s_j}^{\mathrm{NN}}$ to $\acute{f}_{s_j}^{\mathrm{NN}}$ by changing $n$ to $n' = 2mn + n - m$ such that $\delta' = \frac{1}{2n'-1} = \frac{\delta}{M}$. Next, we divide the refined basis functions $\acute{f}_{s_j}^{\mathrm{NN}}$ into $M$ groups, and organize them following the configuration of $f_{s_j}^{\mathrm{NN}}$ along with the required shift. Then we have $M$ pseudo-copies of $f_{s_j}^{\mathrm{NN}}(x)$ denoted by $\acute{f}_{s_j,l}^{\mathrm{NN}}(x), l = 1, 2, .., M$. These pseudo-copies have the same height and their summation approximates $f_{s_j}(x)$ as well as $Mf_{s_j}^{\mathrm{NN}}(x)$. Similarly, we can also choose a sufficiently large $n$ to ensure that

$$E'_u = \max_{x \in [0,1]} \left| \sum_{k' \in K'/K'_{\mathrm{used}}} [p_{k'}\phi_{k'}^+(x) + q_{k'}\phi_{k'}^-(x)] \right| \leq \frac{\Delta h}{2}.$$

Consequently, we have

$$|\acute{f}^{\mathrm{NN}}(x) - g(x)| = \left| \sum_{k' \in K'_{\mathrm{used}}} [p_{k'}\phi_{k'}^+(x) + q_{k'}\phi_{k'}^-(x)] + C' - g(x) \right| + E'_u \leq \Delta h, \quad \forall x \in [0,1]. \tag{21}$$

Here the constant $C'$ comes from the construction of the constant function approximators, which can be small enough for a sufficiently wide network. If $C' < \Delta h$, then omitting $C'$ will not affect the approximation result.

*Proof of Theorem 2.* Similar to the proof of Theorem 1, using the refined construction in Eq. (21), we can finish the proof. □

## H    Estimating the approximation rate in Section 3.4

This section gives a detailed estimation of the step function approximation error in Eq. (8). The piecewise form of step function approximator $f_s^{\mathrm{NN}}$ in Eq. (9) enables us to subdivide the error into four parts like

$$\begin{aligned} e_s^2 &= \left\| \left( f_s^{\mathrm{NN}} + \frac{h}{2} \right) - f_s \right\|_{L^2}^2 = \int_0^1 \left| \left[ f_s^{\mathrm{NN}}(x) + \frac{h}{2} \right] - f_s(x) \right|^2 \mathrm{d}x \\ &= \int_{b_1}^s \left| f_s^{\mathrm{NN}}(x) + \frac{h}{2} \right|^2 \mathrm{d}x + \int_s^{b_4} \left| f_s^{\mathrm{NN}}(x) - \frac{h}{2} \right|^2 \mathrm{d}x \\ &= \mathcal{A} + \mathcal{B} + \mathcal{C} + \mathcal{D}, \end{aligned} \tag{22}$$

where

$$\mathcal{A} = \int_{b_1}^{b_2} \left| (-b_1 + b_4)x + b_1^2 + b_1 b_4 - 2b_2 b_3 - (2b_1 b_4 - 2b_2 b_3) \right|^2 \mathrm{d}x,$$

$$\mathcal{B} = \int_{b_2}^s \left| (-2b_1 + 2b_2)x + b_1^2 - b_2^2 + b_1 b_4 - b_2 b_3 - (2b_1 b_4 - 2b_2 b_3) \right|^2 \mathrm{d}x,$$

$$\mathcal{C} = \int_s^{b_3} \left| (-2b_1 + 2b_2)x + b_1^2 - b_2^2 + b_1 b_4 - b_2 b_3 + (2b_1 b_4 - 2b_2 b_3) \right|^2 \mathrm{d}x,$$

$$\mathcal{D} = \int_{b_3}^{b_4} \left| (-b_1 + b_4)x + b_1^2 - b_2^2 - b_3^2 + b_1 b_4 - (b_1^2 - b_2^2 - b_3^2 + b_4^2) \right|^2 \mathrm{d}x.$$

Before calculating the integral in (22) separately, beware that the $\{b_i\}_{i=1}^4$ are chosen to be symmetrical on both sides of the jump point $x = s$ as

$$b_1 = s - k_2, \quad b_3 = s + k_1, \tag{23}$$
$$b_2 = s - k_1, \quad b_4 = s + k_2, \tag{24}$$

which makes the integral range of error also symmetrical about the jump point (*i.e.*, $\mathcal{A} = \mathcal{D}$ and $\mathcal{B} = \mathcal{C}$). So we only need to calculate the first two parts, which give

$$\begin{aligned}
\mathcal{A} &= \int_{b_1}^{b_2} \left| (-b_1 + b_4)x + b_1^2 - b_1 b_4 \right|^2 \mathrm{d}x \\
&= \left. \frac{\left[ (-b_1 + b_4)x + b_1^2 - b_1 b_4 \right]^3}{3(-b_1 + b_4)} \right|_{b_1}^{b_2} \\
&= -\frac{1}{3}(b_1 - b_2)^3(b_4 - b_1)^2,
\end{aligned}$$

and

$$\begin{aligned}
\mathcal{B} &= \int_{b_2}^{s} \left| (-2b_1 + 2b_2)x + b_1^2 - b_2^2 - b_1 b_4 + b_2 b_3 \right|^2 \mathrm{d}x \\
&= \left. \frac{\left[ (-2b_1 + 2b_2)x + b_1^2 - b_2^2 - b_1 b_4 + b_2 b_3 \right]^3}{3(-2b_1 + 2b_2)} \right|_{b_2}^{s} \\
&= -\frac{1}{6}(b_1 - b_2)^2(b_2 - b_3) \left[ 12b_1^2 + b_2^2 + 4b_2 b_3 + 7b_3^2 - 6b_1(b_2 + 3b_3) \right].
\end{aligned}$$

Thus the error gives

$$e_s^2 = -\frac{1}{3}(b_1 - b_2)^2 \left[ 8b_1^3 - b_2^3 - b_2^2 b_3 + b_2 b_3^2 - 7b_3^3 - 4b_1^2(b_2 + 5b_3) + 4b_1(b_2^2 + 5b_3^2) \right].$$

By applying the relation in Eq. (23) and considering $d = k_2 - k_1$, we get the form in Eq. (8).

Next, we estimate the error of our pseudo-copy $\acute{f}_{s_l}^{\mathrm{NN}}$. The triangle inequality is adopted to estimate the overall approximation error of these stacked $\{\acute{f}_{s_l}^{\mathrm{NN}}\}_{l=1}^M$ to a predetermined step function, *i.e.*,

$$e_s = \left\| \left( \sum_{l=1}^M \acute{f}_{s_l}^{\mathrm{NN}} + \frac{h}{2} \right) - f_s \right\|_2 \leq \sum_{l=1}^M \left\| \left( \acute{f}_{s_l}^{\mathrm{NN}} + \frac{h}{2M} \right) - \frac{1}{M} f_s \right\|_2 =: \sum_{l=1}^M e_{s_l}. \tag{25}$$

Now we focus on the approximation error $e_{s_l}$ of each $f_{s_l}^{\mathrm{NN}}$ to the $f_s/M$. However, the result in Eq. (8) cannot be directly adopted since it only holds for the symmetry case (23), which is unlikely to be satisfied by each $f_{s_l}^{\mathrm{NN}}$ due to uniquely utilized biases. Instead, we can choose biases that are *nearly* symmetrical, with some mismatch measured by $\Delta s$. Therefore, the relation of the mismatched biases $\{\tilde{b}_i\}_{i=1}^4$ in Eq. (23) is transformed into

$$\begin{aligned}
\tilde{b}_1 &= (s + \Delta s) - k_2, \quad \tilde{b}_3 = (s + \Delta s) + k_1, \\
\tilde{b}_2 &= (s + \Delta s) - k_1, \quad \tilde{b}_4 = (s + \Delta s) + k_2.
\end{aligned} \tag{26}$$

Based on Eq. (9), it's clear that this transformation is equivalent to shifting $f_s^{\mathrm{NN}}(x)$ to $f_s^{\mathrm{NN}}(x - \Delta s)$. Consequently, each $e_{s_l}$ in Eq. (25) is estimated by a similar form used in Eq. (22), given by

$$\begin{aligned}
e_{s_i}^2 &= \int_{-1}^1 \left| \left[ f_{s_i}^{\mathrm{NN}}(x - \Delta s) + \frac{h}{2M} \right] - \frac{1}{M} f_s(x) \right|^2 \mathrm{d}x \\
&= \int_{b_1 + \Delta s}^{s} \left| f_{s_i}^{\mathrm{NN}}(x - \Delta s) + \frac{h}{2M} \right|^2 \mathrm{d}x + \int_{s}^{b_4 + \Delta s} \left| f_{s_i}^{\mathrm{NN}}(x - \Delta s) - \frac{h}{2M} \right|^2 \mathrm{d}x,
\end{aligned} \tag{27}$$

where the integral range $[b_1 + \Delta s,\ b_4 + \Delta s]$ is not symmetrical about $x = s$ due to the mismatch. However, since $\Delta s$ is small, we can assume that $b_2 + \Delta s \le s \le b_3 + \Delta s$, then follow the similar procedure as used in Eq. (22) to divide the error in Eq. (27) into four parts. Thanks to the integration by substitution, the integral over $[b_1 + \Delta s,\ b_2 + \Delta s]$ and $[b_3 + \Delta s,\ b_4 + \Delta s]$ is identical to the first and last part $\mathcal{A}$ and $\mathcal{D}$ in Eq. (22), respectively. Therefore, we only need to evaluate the integral over the middle interval $[b_2 + \Delta s,\ b_3 + \Delta s]$. Considering the relationship (23) and $d = k_2 - k_1$, we obtain the following expression

$$
\begin{aligned}
e_{s_i}^2 =& \mathcal{A} + \int_{b_2+\Delta s}^{s} \left| (-2b_1 + 2b_2)(x - \Delta s) + b_1^2 - b_2^2 - b_1 b_4 + b_2 b_3 \right|^2 \mathrm{d}x \\
& + \int_{b_2+\Delta s}^{s} \left| (-2b_1 + 2b_2)(x - \Delta s) + b_3^2 - b_4^2 + b_1 b_4 - b_2 b_3 \right|^2 \mathrm{d}x + \mathcal{D} \\
=& \frac{8}{3}(k_1 - k_2)^2 \left[ k_1^3 + 3k_1^2 k_2 + 2k_1 k_2^2 + k_2^3 + 3\Delta s^2 (k_1 + k_2) \right] \\
=& -\frac{8}{3} d^2 \left( d^3 - 6d^2 k_2 + 11dk_2^2 - 7k_2^3 + 3d\,\Delta s^2 - 6k_2 \Delta s^2 \right).
\end{aligned}
$$

Since $\Delta s$ can be assumed to be small as $\mathcal{O}(d)$, we can obtain an estimation similar to Eq. (8), where $e_{s_i} \sim \mathcal{O}(d^{5/2})$.

## I   DETAILED PROOF OF THEOREM 3

In the extension of our equidistant situation to the commonly used random initializations, the basic proof ideas remain unchanged, but the error introduced by randomness must be carefully controlled. The estimation of the equidistant case is based on the symmetry construction of the step function approximator, which is invalid with random initializations because we can't always locate the desired basis function accurately. However, as the width increases, the randomly sampled basis functions will become denser, thus approaching the equidistance case. Therefore, we can argue that for a sufficiently wide network, it is highly possible to find a basis function that closely matches the location required for the construction of step function approximators.

The proof begins with applying Theorem 1, which indicates that for any error threshold $\varepsilon$ and continuous function $f^*$, there is an equidistantly initialized network $f^{\mathrm{NN}}$ in Eq. (2) with parameters $B_*^{(n)} = (b_i)_{i=1}^n$ and $W_*^{(n)} = (\pm b_i)_{i=1}^n$, such that

$$
\| f^{\mathrm{NN}} - f^* \|_{L^\infty} < \varepsilon/6.
$$

Next, for a sufficiently wide network initialized randomly from uniform distributions, we can find a subnetwork $f_{\mathrm{sub}}^{\mathrm{NN}}$ with parameters $\{B_r^{(n)}, W_r^{(n)}\}$, which can be viewed as randomly perturbed from $f^{\mathrm{NN}}$ as

$$
B_r^{(n)} = (b_i + r_i^B)_{i=1}^n, \quad W_r^{(n)} = \left( \pm (b_i + r_i^W) \right)_{i=1}^n,
$$

where $r_i^B \sim \mathcal{U}[-\Delta r_B, \Delta r_B]^n$, $r_i^W \sim \mathcal{U}[-\Delta r_W, \Delta r_W]^{2n}$ measured the perturbation, and $\Delta r_B$, $\Delta r_W$ are the maximum allowable disturbance range of $B_*^{(n)}$ and $W_*^{(n)}$, respectively. Consequently, for sufficiently small $\Delta r_W$, $\Delta r_B$, the subnetwork will also process the approximation power, and the approximation error $E_{s,r}$ gives

$$
E_{s,r} := \| f_{\mathrm{sub}}^{\mathrm{NN}} - f^* \|_{L^\infty} < \varepsilon/3. \tag{28}
$$

Therefore, for the case $\Delta r := \Delta r_B = \Delta r_B$, the probability of finding at least one basis function in the interval $[b_i - \Delta r, b_i + \Delta r]$ is $1 - (1 - 2\Delta r)^{n/4J}$, because for each step function, we have $n/J$ uniformly distributed basis functions, where $J$ is the number of step function that needs to be approximated. It also holds for the probability of coefficients $W_r^{(n)}$. Thus, under the probability of $[1 - (1 - 2\Delta r)^{n/4J}]^8$, we can construct a similar step function approximator $f_{s,r}^{\mathrm{NN}}$ like Eq. (3) using the basis functions located at $\{b_i + r_i\}_{i=1}^4$. Hence the approximator of the target function $f^*$ can be achieved with the probability of

$$
\mathbb{P}\left[ E_{s,r} < \varepsilon/3 \right] = [1 - (1 - 2\Delta r)^{n/4J}]^{8J} \le 1 - (1 - 2\Delta r)^{2n}. \tag{29}
$$

To be consistent with the setting in Theorem 3, where $B_r^{(n)} \sim \mathcal{U}[0,1]^n$ and $W_r^{(n)} = (\pm p_i)_{i=1}^n$, $p_i \sim \mathcal{U}[0,1]$, we can impose further constraints on the parameters near the boundary of $[0,1]$ to prevent them from exceeding the range of $[0,1]$. The formal proof is given as follows.

*Proof of Theorem 3.* For any $\varepsilon > 0$ and $f^* \in C[0,1]$, we can choose a small $\Delta r = \bar{\varepsilon}$ to ensure that

1. $E_{s,r} < \varepsilon/3$ based on Eq. (28);

2. $\mathcal{I} < \varepsilon/3$ based on Eq. (17).

Then let $n$ sufficiently large to enable that

1. $\mathbb{P}\Big[E_{s,r} < \varepsilon/3\Big] \geq 1 - \delta$ based on Eq. (29);

2. $\mathcal{J} < \varepsilon/3$ based on Eq. (17).

Hence, the overall approximation error of our network $f_r^{\text{NN}}$ with random initialization to the target function $f^*$ can be

$$|f_r^{\text{NN}}(x) - f^*(x)| \leq E_{s,r} + E_{u,r} \leq E_{s,r} + \mathcal{I} + \mathcal{J} < \varepsilon, \quad x \in [0,1]. \tag{30}$$

Hence we can finish the proof. □

## J   ADAM-BASED LAPERM ALGORITHM

Here we briefly introduce the Original Adam-based LaPerm algorithm with a fixed permutation period $k$ in Qiu & Suda (2020). The pseudocode is shown in Algorithm 1.

---
**Algorithm 1** Original Adam-based LaPerm algorithm
---
**Require:** Loss function $L$, initial weights $W$, training set $D_T$
**Require:** Permutation period $k$, Maximum training epoch $M_e$
**Require:** Inner optimizer Adam
  $\theta_0 = W$    // Initialize the weights
  **for** $t = 1, 2, \ldots, M_e$ **do**
    $\theta_t \leftarrow \text{Adam}(L, \theta_{t-1}, D_T)$    // Free training by Adam
    **if** $k$ divides $t$ **then**
      $\theta_t \leftarrow \tau_t(W)$    // Apply the permutation
    **end if**
  **end for**

---

This algorithm rearranges the initial weights $W$ guided by the order relationship of $\theta_t$, so the trained weights will hold the initial value for all $t = 1, 2, \ldots, M_e$. Therefore, it can be regarded as a permutation of the initial weights $W$.

Moreover, we propose an alternative permutation algorithm based on a simple intuition. Since the permutation requires the order relationships encoded by the training process, the permutation should be applied after learning a certain amount of information. To achieve this, we gradually increase the permutation period $k$ during training to match it with the slower weight update induced by a decaying learning rate of the optimizer. It will guarantee that sufficient information is learned to guide the next permutation.

In this way, we propose a relaxed LaPerm algorithm, which allows for an adjustable $k$ that exponentially increases during the training process with a coefficient $\gamma_k$. The pseudocode for this algorithm is shown in Algorithm 2.

---

**Algorithm 2** Relaxed LaPerm algorithm

---

**Require:** Loss function $L$, initial weights $W$, training set $D_T$
**Require:** Permutation period $k$, Maximum training epoch $M_e$, exponential decay coefficient $\gamma_k$
**Require:** Inner optimizer Adam

  $\theta_0 = W$    // Initialize the weights
  **for** $t = 1, 2, \ldots, M_e$ **do**
    $\theta_t \leftarrow \text{Adam}(L, \theta_{t-1}, D_T)$    // Free training by Adam
    **if** $k$ divides $t$ **then**
      $\theta_t \leftarrow \tau_t(W)$    // Apply the permutation
      $k \leftarrow k/(\gamma_k)^t$    // Adjust the period $k$
    **end if**
  **end for**

---

## K    APPROXIMATING THE TWO-DIMENSIONAL CONTINUOUS FUNCTIONS

As a natural extension, we consider a two-dimensional functions $z = -\sin \pi xy$, where $(x, y) \in [-1, 1] \times [-1, 1]$, starting with the construction of basis functions like $\phi_i^{\pm}(x)$ in Eq. (1). Recall that in the one-dimensional case, the two subsets of basis $\phi_i^{\pm}(x)$ correspond the two opposite directions along the $x$-axis. Therefore, at least four directions are required to represent a function defined on the $xy$-plane, of which two are parallel to the $x$-axis as $\phi_i^{\pm}(x, \cdot) = \text{ReLU}(\pm(x-b_i))$ and $\phi_j^{\pm}(\cdot, y) = \text{ReLU}(\pm(y-b_j))$ for $y$-axis, respectively. Furthermore, inspired by the lattice Boltzmann method in fluid mechanics (Chen & Doolen, 1998), we introduce another four directions as $\psi_k^{\pm\pm}(x, y) = \text{ReLU}(\pm x \pm y - b_k)$. So the whole basis functions are divided into eight subsets, each corresponding to a different direction (see Fig. 5(b)). Also, the range of biases is essential since the distribution of $\psi_k^{\pm\pm}(x, y)$ must be at least $\sqrt{2}$-times wider to cover the entire domain. Here we set the biases to range in varying directions with a uniform scaling factor, providing flexibility in dealing with the unwanted coefficients.

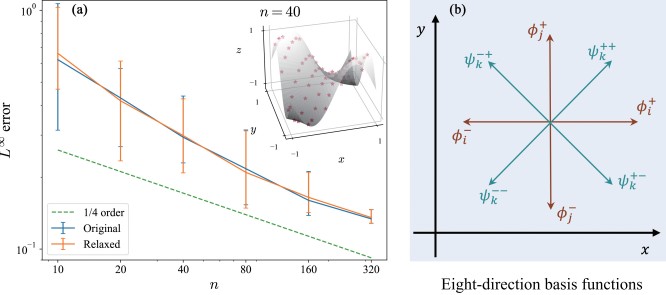

Figure 5: (a) Approximating two-dimensional continuous function $z = -\sin \pi xy$, where $x, y \in [-1, 1] \times [-1, 1]$. The inset panel presents the target function surface and an example of the approximation result as dots. (b) The two-dimensional basis function settings.

Accordingly, we utilize a 2-8$n$-1-1 network architecture and follow the same setting as before (refer to App. L). The results depicted in Fig. 5(a) also show good approximation power. However, the 1/2 convergence rate in previous cases cannot be attained here. We hypothesize that this is due to our preliminary eight-direction setting of the basis functions. This degeneration indicates the challenge of extending our theoretical results to higher dimensions. Further research will address this difficulty by considering more appropriate high-dimensional basis function setups. One possible solution is utilizing ReLU-generalized multi-variable activation functions such as the Maxout activation function Goodfellow et al. (2013), which would provide more flexibility to our approximator construction. Another promising way is allowing the fixed weights in the first hidden layer to be permuted, which is not applicable for scalar input but may be helpful in high-dimensional cases.

Moreover, as discussed in Sect. 4.4, the mismatch between the existing implementations and the permutation setting poses numerical challenges of permutation training in higher dimensions. The

performances are significantly affected by the algorithm implementations and initialization settings, both of which need further investigation and are beyond the scope of this work. We hope our work can inspire and motivate the development of more sophisticated implementations specific to permutation training.

The permutation training, as a numerical algorithm, can be directly applied to high-dimensional cases, even if it requires a significantly larger network width. Using a similar numerical setting, we can also approximate functions with three-dimensional inputs. Here we consider $f(x, y, z) = \sin 3x \cdot \cos y \cdot \sin 2z$, where $(x, y, z) \in [-1, 1]^3$. The results plotted in Fig. 6 demonstrate a certain degree of approximation power. However, a degeneration convergence rate from $1/2$ to $1/6$ also indicates the theoretical limitations of the current construction.

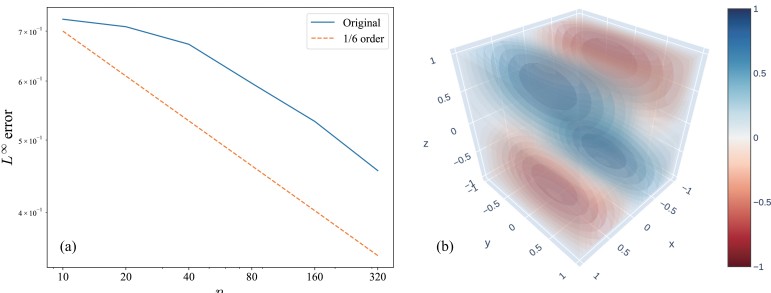

Figure 6: Approximating three-dimensional continuous function $f(x, y, z) = \sin 3x \cdot \cos y \cdot \sin 2z$, where $(x, y, z) \in [-1, 1]^3$. (a) The convergence behavior under random seed 2022. (b) The three-dimensional illustration of the target function, where the function value $f(x, y, z)$ is plotted by the corresponding color in the color bar.

## L    THE EXPERIMENTAL SETTING

To establish the convergence property upon increasing the width of the network, we sample the training points randomly and uniformly in $[-1, 1]$, along with equidistantly distributed test points. The maximum training epoch is sufficiently large to ensure reaching the stable state. For the two-dimensional case, we set the basis functions at a larger domain than the functions to ensure accuracy near the boundary. The scale is measured by $T_b$, which means the biases are in $[-1 - T_b, 1 + T_b]$ in each dimension. The detailed choice is in Table 1.

Table 1: Hyperparameters setting.

| Hyperparameters | 1D |
|---|---|
| Architectures | 1-2$n$-1-1 |
| $k$ | 5 |
| Batch size | 16 |
| # training points | 1600 |
| # test points | 400 |
| $T_b$ | 0 |
| $n$ | $\{10, 20, 40, 80, 160, 320\}$ |
| # epoch | 6400 |
| Learning rate (LR) | 1e-3 |
| Multiplicative factor of LR decay | 0.998 |
| Multiplicative factor of $k$ increase | $\sqrt[10]{1.002}$ |

The experiments are conducted in NVIDIA A100 Tensor Core GPU with a 40GB PCIe interface. However, our code is hardware-friendly since each case only consumes approximately 2GB of memory. The code implementation uses *torch.multiprocessing* in PyTorch 2.0.0 with ten different random seeds, namely $2022, 3022, \cdots, 12022$. Additionally, the training data of each case is sampled under

the random seed 2022 to ensure that they are comparable. The choice of hyperparameters in the multi-dimensional cases is listed in Table 2.

Table 2: Hyperparameters setting of multi-dimensional cases.

| Hyperparameters | Dimensions | |
| --- | --- | --- |
| | 2D | 3D |
| Architectures | 2-8$n$-1-1 | 3-26$n$-1-1 |
| $k$ | 5 | 20 |
| Batch size | 128 | 640 |
| $T_b$ | 0.75 | |
| # training points | 51200 | |
| # test points | 12800 | |
| $n$ | $\{10, 20, 40, 80, 160, 320\}$ | |
| # epoch | 6400 | |
| Learning rate (LR) | 1e-3 | |
| Multiplicative factor of LR decay | 0.998 | |
| Multiplicative factor of $k$ increase | $\sqrt[10]{1.002}$ | |

## M    SAMPLE OF PERMUTED NETWORKS

Fig. 7 gives an example of the represented functions with different permutations, which provides a first impression of the output functions of permuted networks. Here the weights $W^{(n)}$ in the second hidden layer will be initialized in various ways.

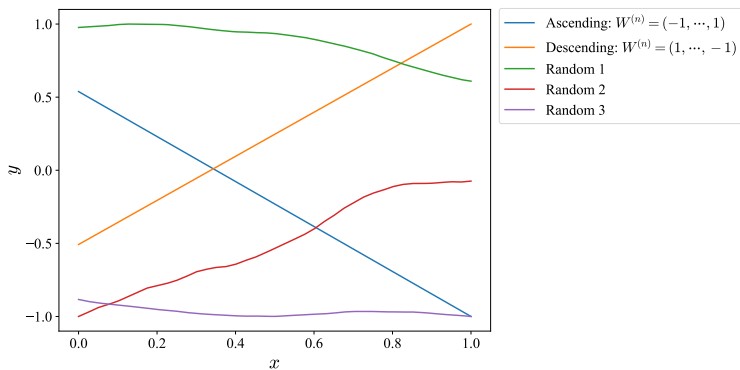

Figure 7: The initial function $f_s^{\mathrm{NN}}$ with different initial order in $W^{(n)}$, whose order is arranged as ascending, descending, and random using different random seeds $2022, 2046, 4096$, respectively.

Fig. 8 gives an example of the step function approximator $f_s^{\mathrm{NN}}$ of a target function $f_s$.

## N    THE IMPACT OF PERMUTATION PERIOD $k$'S VALUE ON CONVERGENCE BEHAVIOR

As a hyperparameter, the choice of permutation period $k$ during the implementation of LaPerm algorithms has the possibility to affect the convergence behavior. Qiu & Suda (2020) reported an unambiguous correlation between the value of $k$ and the final accuracy (refer to Fig. 6 in Qiu & Suda (2020)). Generally, a larger $k$ is associated with slightly higher accuracy of single permutation training result, thus in our experiments, the weights are permuted after each $k$ epoch. Fig. 9 evaluates the impact of $k$'s value on convergence behavior, whose results suggest that this effect remains negligible.

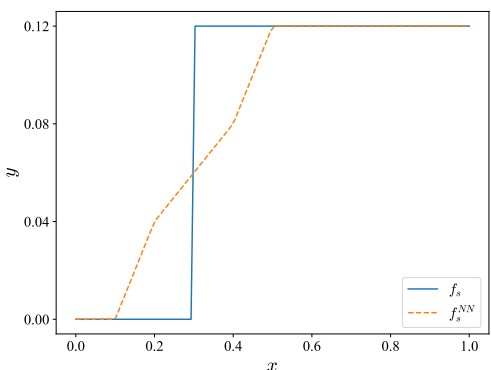

Figure 8: Approximating the step function $f_s$ at $s = 0.3$ with a height $h = 0.12$ by $f_s^{\text{NN}}$ in Eq. (3), where $b_1 = 0.1$, $b_2 = 0.2$, $b_3 = 0.4$, $b_4 = 0.5$.

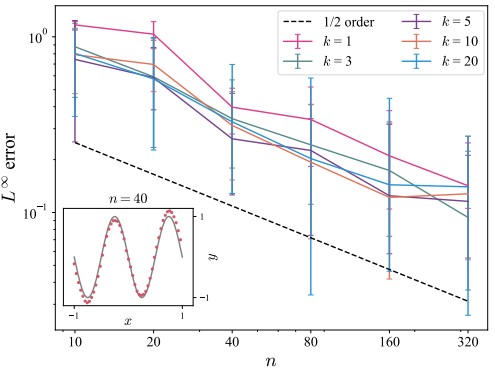

Figure 9: Approximating one-dimensional continuous function $y = a_0 + a_1 \sin(\pi x) + a_2 \cos(2\pi x) + a_3 \sin(3\pi x)$ with equidistantly initialized network, where $x \in [-1, 1]$, and the value of permutation period $k = 1, 3, 5, 10, 20$, respectively. The inset in each panel presents the target function as lines and an example of the approximation result as dots.

## O   GENERALIZE TO THE NETWORKS EQUIPPED WITH LEAKY-RELU

Extending our outcomes to leaky-ReLU can be straightforward. This is because of the two crucial techniques deployed in our proof, specifically, constructing the step function approximators and eliminating the unused parameters, both of which can be readily applied to the leaky-ReLU.

Since our two-direction setting of basis function $\phi^\pm$ in Eq. (1) can impart leaky-ReLU with symmetry equivalent to ReLU, it's feasible to construct a similar step function approximator by rederiving the relevant coefficient $p_i, q_i$. Furthermore, the existing eliminating method can be directly employed for leaky-ReLU, due to the capacity to reorganize a pair of leaky-ReLU basis functions into linear functions for further processing.

As an initial attempt, we numerically examine the networks equipped with leaky-ReLU by only changing the activation function in the case of Fig. 2(a). The results plotted in Fig. 10 provide clear evidence of the approximation power of networks equipped with leaky-ReLU.

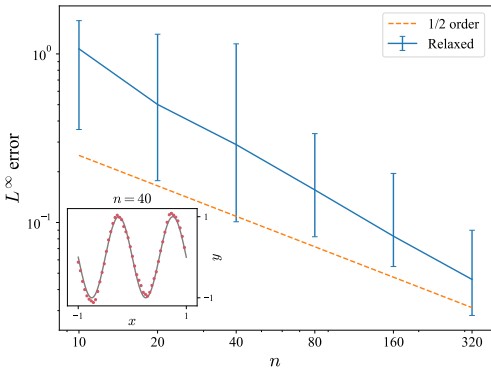

Figure 10: Approximating one-dimensional continuous function $y = -\sin(2\pi x)$ with equidistantly initialized network equipped with leaky-ReLU, where $x \in [-1, 1]$. The inset in each panel presents the target function as lines and an example of the approximation result as dots.

Nonetheless, the approximation based on leaky-ReLU may result in different constants during the proof, leading to potential discrepancies in details like the approximation order when compared with the ReLU-based conclusions.

## P   EXTENDING THE RESULTS TO MORE COMPLICATED ARCHITECTURES

Here we discuss how to extend our UAP result to the networks with deeper architecture, residual connections, or sparse initializations. Our work follows the conventional setting of free training UAP study and primarily focuses on shallow fully connected networks. This is based on the premise that the conclusions can be readily extended to scenarios involving deeper networks or those incorporating residual connections.

As for deep networks, the application of permutation training may inhibit a direct extension of the previous methodology. However, our approach facilitates a direct implementation within deeper networks. Note that we can construct an identity function $y = x$ using a pair of basis functions $y = p_i\phi_i^+(x) + q_i\phi_i^-(x)$, where $b_i = 0, p_i = 1, q_i = -1$. This process enables us to utilize identity functions within subsequent layers by adding a certain pair of parameters. Consequently, the deep networks scenario parallels the cases discussed in this paper, which allows us to realize UAP within deep networks.

The case of networks with residual connections can be addressed following the conventional way. Denoting a residual block as $x_{k+1} = F(x_k) + x_k$, where $x_k$ and $x_{k+1}$ are the inputs and outputs, respectively, along with the fully connected subnetwork $F$. The problem can be transfigured into learning $x_{k+1} - x_k$, the difference between inputs and outputs, with a network $F$, thereby can be solved by our proved UAP results.

Additionally, this paper's conclusions could be extended to sparse initialization scenarios, an area where permutation training shows significant advantages (see Fig. 6(e) in Qiu & Suda (2020)). In our future research, we aim to theoretically depict this immense potential. On an intuitive level, our proof can explain the advantages of permutation training in sparsity scenarios. Notice that a primary challenge of permutation training is managing unused basis functions. However, in sparse situations, we can conveniently deal with these remaining basis functions simply by assigning them a zero coefficient, thereby facilitating the approximation process.

## Q    SUPPLEMENTARY FIGURES FOR ONE-DIMENSIONAL CASES

Apart from the pairwise random initializations presented in Fig. 2, we also conduct the experiments under the equidistant scenario related to Theorem 1. The results are plotted in Fig. 11, and the reference dash lines are in the same position as in Fig. 2 for the convenience of comparison. It can be found that the randomly initialized cases exhibit some performance advantages, especially in the convergence rate.

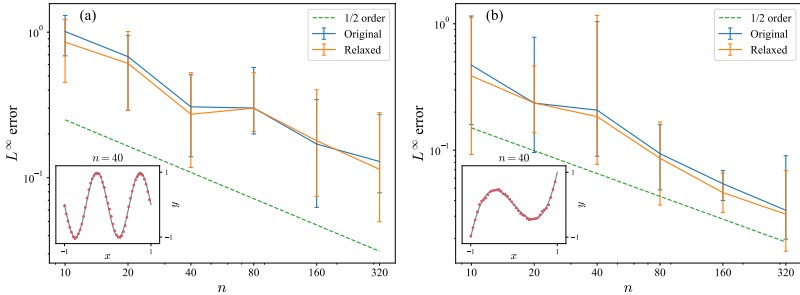

Figure 11: Approximating one-dimensional continuous function (a): $y = -\sin(2\pi x)$ and (b): $y = \frac{1}{2}(5x^3 - 3x)$ with equidistantly initialized network, where $x \in [-1, 1]$. The inset in each panel presents the target function as lines and an example of the approximation result as dots.

Next, we consider a more complicated example with some multiscale behaviors by choosing a fourth-order Fourier series with random coefficients like

$$y = a_0 + a_1 \sin(\pi x) + a_2 \cos(2\pi x) + a_3 \sin(3\pi x), \quad x \in [-1, 1], \tag{31}$$

where the coefficients are randomly chosen in $[0.1]$ as $\{0.3958, 0.9219, 0.7588, 0.3811\}$ from the random seed *2022*. The approximation results are plotted in Fig. 12, which again shows the approximation ability of the complicated continuous function. The hyperparameters are set as Table 3, where the values highlighted in bold are chosen to be different from the case in Table 1.

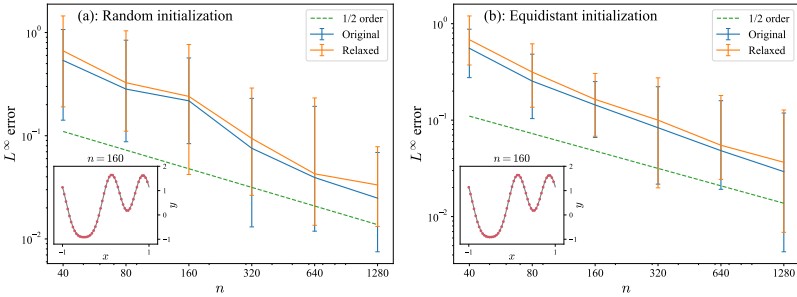

Figure 12: Approximating one-dimensional continuous function $y = a_0 + a_1 \sin(\pi x) + a_2 \cos(2\pi x) + a_3 \sin(3\pi x)$ with (a): pairwisely randomly initialized and (b): equidistantly initialized network, where $x \in [-1, 1]$. The inset in each panel presents the target function as lines and an example of the approximation result as dots.

Table 3: Hyperparameters setting of Fig. 12.

| Hyperparameters | |
|---|---|
| Architectures | 1-$2n$-1-1 |
| $k$ | **20** |
| Batch size | 16 |
| # training points | **12800** |
| # test points | **3200** |
| $T_b$ | 0 |
| $n$ | **\{40, 80, 160, 320, 640, 1280\}** |
| # epoch | 6400 |
| Learning rate (LR) | 1e-3 |
| Multiplicative factor of LR decay | 0.998 |
| Multiplicative factor of $k$ increase | $\sqrt[10]{1.002}$ |

