# OpenReview forum: "Neural Networks Trained by Weight Permutation are Universal Approximators"
_ICLR.cc/2024/Conference — Submitted to ICLR 2024_

### Official Review · Reviewer_5HxL · 2023-10-27

**Soundness:** 3 good
**Presentation:** 3 good
**Contribution:** 2 fair
**Rating:** 5
**Confidence:** 4

**Summary:**

This paper demonstrates that the permutation training technique can effectively steer a ReLU network to approximate one-dimensional continuous functions, effectively realizing the universal approximation property in the context of the permutation training method. Then they empirically confirm their theoretical results.

**Strengths:**

Clarity
- The paper is written effectively and ensures high accessibility.
- Theorem and proofs sketch are simple and easy to follow.

Originality
- This paper theoretically and empirically shows the universal approximation property of the permutation training method for the 1d regression case.

**Weaknesses:**

Main Results
- While this paper theoretically showcases the effectiveness of the permutation training method in the context of one-dimensional regression, the simplicity of this result may not fully validate the method's performance for larger and more complex deep learning models.

Experiments
- They conducted too simple experiments.

**Questions:**

Experiments
- It would be beneficial if the paper included empirical experiments demonstrating the performance of the permutation training method on high-dimensional regression or classification tasks, even without the inclusion of precise theoretical results.
- Their newly introduced relaxed LaPerm algorithm doesn't appear to offer any significant advantages over the original LaPerm. Hence, I believe the authors should present the results of other proposed algorithms, such as those employing a self-adjusted strategy.

---

> ### Author Response · Authors · 2023-11-20
> **Response to Reviewer 5HxL (pt 1/2)**
>
> We thank the reviewer for the constructive comments. We hope to have addressed all the issues by the reviewer in our revised manuscript and in our replies below.
>
> 1. **More complicated experiments.** We understand the reviewer's concern from a numerical perspective. However, we believe that the main contribution of this paper lies in its theoretical aspect. The purpose of presenting numerical experiments in our paper is solely to support and validate the theoretical proof. Therefore, our numerical experiments are conducted to be consistent with the theoretical setting. *We have rewritten the advance organizer of Sect. 4 to clarify this purpose (highlighted in red in the first paragraph of Sect. 4 on Page 6).*
>    - Our work provides a theoretical foundation for permutation training by proving it possesses UAP. To the best of our knowledge, this is the first theoretical result regarding the permutation training setting, which paves the way for further investigation. As for the experimental part, in fact, it is uncommon in theoretical papers discussing UAP to conduct empirical experiments. Nevertheless, since permutation training remains relatively uncommon, we want to enhance readers' understanding of our theoretical results through some intuitive numerical outcomes.
>    - As for the large-scale high-dimensional classification tasks, Qiu & Suda (2020) have conducted extensive numerical experiments. The experimental scale covered small-scale CNNs trained on the MNIST dataset, as well as ResNet50 and VGG-based networks on the CIFAR-10 dataset (see Table 1 in Qiu & Suda (2020)). Based on the reported numerical evidence, we believe that the permutation training has adequately demonstrated its effectiveness in image classification tasks.
>    - Applying permutation training on high-dimensional regression tasks had not been numerically explored, probably due to the absence of representative high-dimensional datasets and the challenges in visualizing the fitting results. Given the theoretical nature of this paper, we didn't allocate too much computational resources to conducting high-dimensional experiments. Our initial attempt in the 2D case (see App. K) also aims to examine the capacity of our proof in multi-dimensional scenarios. To be consistent with our theoretical construction, we manually organize and fix the weights of the first hidden layer, which may impose external limitations. However, we believe that permutation training, as a numerical algorithm, can be directly applied to high-dimensional cases, even if it requires a significantly larger network width.
>    - *As a response to the reviewer's feedback, we are endeavoring to numerically consider an additional 3D function approximation example, and the preliminary results are presented in the newly added Fig. 6 in App. K. The existing results also demonstrate the approximation power. However, due to time and resources constraints during the rebuttal process, we are unable to present a comprehensive numerical case. If this paper is accepted, we will make every effort to refine this case in the final version.*

---

> ### Author Response · Authors · 2023-11-20
> **Response to Reviewer 5HxL (pt 2/2)**
>
> 2. **The results of other algorithms.** We acknowledge that the proposed relaxed LaPerm algorithm doesn't show significant advantages. However, we think proposing other algorithms is beyond the scope of this paper because our main objective is to theoretically explore a novel training method rather than numerically developing some new algorithms.
>    - The basic concept of the relaxed LaPerm algorithm is proposed by Qiu & Suda (2020), which suggests the feasibility of adjusting the permutation period $k$ to be consistent with the learning rate decay. To verify this idea, we adopt an exponentially relaxed scheme as an initial attempt. To be consistent with our proof construction, our manual setting may also impose limitations on the algorithm's performance. In other scenarios, the relaxed LaPerm could still be a considerable choice. It's also possible to consider more complicated strategies, such as the self-adjusted scheme with respect to the loss behavior. However, we only suggest this as a possible direction without actually implementing it, because proposing new algorithms is not the main aim of our work.
>    - Nevertheless, we do agree that the existing algorithm may not fully leverage the potential of permutation training. Our work also highlights the necessity for developing more sophisticated algorithms and initializations specifically tailored for permutation training. We hope our constructive proof can inspire and motivate the development of more sophisticated algorithms, enabling this training method to be truly competitive in practical applications. It's also very promising to take the algorithms in other areas as a reference. Moreover, our proof is independent of the algorithm implementation, which can always serve as the theoretical foundation of this training method.

---

> ### Comment · Reviewer_5HxL · 2023-11-21
> **Response to Authors' Response**
>
> Sincerely thank the authors for their thorough explanations. I comprehend the authors' emphasis on the theoretical exploration of UAP in permutation training. However, I maintain the view that relying solely on theoretical findings in a 1D regression task restricts the broader understanding within the community. Hence, I suggested conducting additional experiments to confirm if analogous trends manifest in high-dimensional experiments. I also understand that the newly introduced algorithm is not the main aim of this work. And my rationale for proposing this suggestion aligns with the aim of adding more contributions beyond theoretical findings. Overall, considering this version of the paper, my evaluation score will remain unchanged.

---

### Official Review · Reviewer_XdGX · 2023-10-30

**Soundness:** 3 good
**Presentation:** 3 good
**Contribution:** 3 good
**Rating:** 6
**Confidence:** 3

**Summary:**

The authors study the UAP (universal approximation problem) of deep neural networks. They derive that some permutation-trained networks could achieve UAP. First, they show that it is true if the parameters are selected as $\frac{i}{n-1} (0\leq i \leq n-1)$. Secondly, they generalized their results to the scenario with random initialization.

**Strengths:**

Originality: The related works are adequately cited. This paper derives that some permutation-trained networks with parameters selected from $\frac{i}{n-1} (0\leq i \leq n-1)$ could achieve UAP. Furthermore, the authors generalized their results to the DNNs with random initialization. The main results in this paper will certainly help us have a better understanding of the universal approximation property of deep neural networks from a theoretical way. I have checked the technique parts and found that the proofs are solid.

Quality: This paper is technically sound.

Clarity: This paper is clearly written. I find it is easy to follow.

Significance: I think the results in this paper are significant, as explained above.

**Weaknesses:**

For the weights $\frac{i}{n-1} (0\leq i \leq n-1)$, the UAP is always true. For random initialization, the UAP is true with some high probability. It would be interesting to find out for which set of parameters, the UAP is always true. More explanations about this should be addressed.

**Questions:**

More explanations about when UAP is always true should be addressed (for which sets of parameters?). It would also be interesting to derive the results for more activation functions and more architectures used in practice.

---

> ### Author Response · Authors · 2023-11-20
> **Response to Reviewer XdGX (pt 1/2)**
>
> We thank the reviewer for the constructive comments. We are encouraged that the reviewer found our result helpful for a better understanding. We hope to have addressed all the issues by the reviewer in our revised manuscript and in our replies below.
>
> 1. **For which sets of parameters, the UAP is always true?** We appreciate the reviewer for the insightful comments. We also believe that this issue deserves further exploration. Since permutation training doesn't alter the specific value of weights, the role of initialization becomes particularly important. Moreover, as noted in Sect. 4.4, traditional initialization methods underperform in the context of permutation training. Therefore, evaluating and guiding the initialization methods under the permutation training scenario is a problem of substantial value. The systematic characterization of initializations that satisfy the UAP poses a significant challenge, primarily due to the specific settings of permutation training. Nevertheless, based on our experience, any initialization that would be satisfactory under permutation training should, at a minimum, adhere to two conditions:
>
>    - The initialized weight range should cover the complete domain. Since permutation training cannot alter the precise value of the weight, the constructed approximator lacks positional flexibility. If the initialization scale is mismatched, the error will become uncontrollable in uncovered areas. Hence, the commonly used $U_H$ and $U_X$, derived from free training, fail to possess approximation qualities.
>    - It's vital for the weight distribution to be as uniform as possible, thereby equipping it with consistent approximation capacity at different positions. This principle also arises from the fact that permutation training doesn't affect weight distribution. We speculate that this may explain why methods like $N_H$, which are based on normal distribution, lack the approximation features.
>
>    *We have added the relevant discussion to the revised version to underscore the importance of initializations (highlighted in red in Lines 1-2 of the first paragraph of Sect. 4.4 on Page 7).* However, given the limitation in paper length, we may leave a comprehensive exploration for future work.

---

> ### Author Response · Authors · 2023-11-20
> **Response to Reviewer XdGX (pt 2/2)**
>
> 2. **More activation functions and more architectures.** We agree that the generalizability of this paper's theory holds significant importance, and are devoted to extending our findings to an extensive variety of scenarios.
>
>    - We believe that extending our outcomes to leaky-ReLU can be straightforward.  *Our newly added numerical experiment also demonstrates the approximation power (see Fig. 10 in App. O).* This is because of the two crucial techniques deployed in our proof, specifically, constructing the step function approximators and eliminating the unused parameters, both of which can be readily applied to the leaky-ReLU.
>
>      - It's feasible to construct a similar step function approximator by rederiving the relevant coefficient $p_i, q_i$, since our two-direction setting of basis function (i.e., $\phi^+$ and $\phi^-$) can impart leaky-ReLU with symmetry equivalent to ReLU.
>      - Furthermore, the existing eliminating method can be directly employed for leaky-ReLU, due to the capacity to reorganize a pair of leaky-ReLU basis functions into linear functions for further processing.
>
>      *We have added the relevant discussion to the revised version (highlighted in red in Lines 1-2 of the first paragraph of Sect. 4.4 on Page 7). along with the majority of corresponding contents are organized in the App. O.*
>
>    - In response to extending our approach to other network architectures, our work follows the conventional setting of free training UAP study and primarily focuses on shallow fully connected networks. This is based on the premise that the conclusions can be readily extended to scenarios involving deeper networks or those incorporating residual connections.
>
>      - As for deep networks, the application of permutation training seems to inhibit a direct extension. However, our approach can facilitate the implementation. Note that we can construct an identity function $y = x$ using a pair of basis functions $y = p_i \phi_i^+(x) +q_i \phi_i^-(x)$, where $b_i = 0$, $p_i = 1$, $q_i = -1$. This process enables us to utilize identity functions within subsequent layers by adding a certain pair of parameters. therefore, the deep networks scenario parallels the cases discussed in this paper, which allows us to realize UAP within deep networks.
>      - The networks with residual connections can be addressed following the conventional way. Denoting a residual block as $x_{k+1} = F(x_k) + x_k$, where $x_k$ and $x_{k+1}$ are the inputs and outputs, respectively, along with the fully connected subnetwork $F$. The problem can be transfigured into learning the difference between inputs and outputs (i.e., $x_{k+1} - x_k$) with a network $F$, thereby can be ensured by our proven UAP results.
>
>      *We have added the relevant discussion to the revised version (highlighted in red in Lines 2-3 of the second paragraph of Sect. 5 on Page 9), along with the majority of corresponding contents organized in the App. P.*

---

### Official Review · Reviewer_BYmb · 2023-11-01

**Soundness:** 3 good
**Presentation:** 3 good
**Contribution:** 2 fair
**Rating:** 5
**Confidence:** 3

**Summary:**

Universal approximation is a desirable property of neural networks. In this paper, permuting weights of a ReLU network is shown to have the universal approximation property, i.e. given a sufficiently wide network one can fit an arbitrary continuous function as closely as desired, only by permuting that network's weights. The proof involves constructing approximations of step functions out of four pairs of weights. Both random and fixed initializations for the network weights are considered.

**Strengths:**

The work pushes forward UAP proof techniques to a novel situation (training with permutation only), and solves this difficult and constrained case. The proof involves some tricky constructions which could inspire other manipulations of ReLU networks. Some intriguing connections are discussed about random initialization for permutation training techniques, and dynamics during permutation training.

**Weaknesses:**

The lack of multidimensional inputs is a pretty big limitation since many non-trivial networks operate on multidimensional inputs. However, the conclusion does point this limitation out, and it is reasonable to expect it as a follow up work.

The pairwise constraint really limits how the proof can be applied to random initialization cases. Random weights usually suggest we cannot control what the weights can be, but if we can make pairs of weights identical, why not just use the fixed initialization scheme instead? In general the proposed proof method seems to be too specific to adapt to other situations - in the case of truly random weights, error in the constant regions of the stepwise approximators could accumulate globally (see questions for a different suggestion). It would be interesting to understand why permutation training fails on some random initialization schemes and not others, however, as that could point to some theoretical or empirical justification for the pairwise constraint.

**Questions:**

Is it possible to relax the pairwise constraint in a bounded input interval (e.g. [0, 1]) by having error cancel out in the same way that the unusued parameters annihilate?

Instead of annihilating the unused parameters, could we simply divide the stepwise approximation of $f^*$ into more steps to use up the remaining parameters?

Could the authors elaborate on what they observe in figure 4? Specifically, how it relates to rank structures in permutation groups (e.g. larger/smaller cycles take longer/shorter to train), and how it relates to weight consolidation/pruning/weight projection. The connections aren't obvious to the reader.

---

> ### Author Response · Authors · 2023-11-20
> **Response to Reviewer BYmb (pt 1/2)**
>
> We thank the reviewer for the constructive comments. We are encouraged that the reviewer found our proof to be inspirable. We hope to have addressed all the issues by the reviewer in our revised manuscript and in our replies below.
>
> 1. **Pairwisely random initialization.** We acknowledge the inherent limitations of the pairwise random initialization compared to the truly random setting. However, the pairwise initialization remains a much closer approximation to practicality than the fixed scenario, primarily due to the added complexity introduced by randomness. We believe that by exploring this scenario, valuable insights and inspiration can be garnered for future research in truly random settings.
>    - We have endeavored to generalize to truly random initializations, which is already numerically demonstrated by our numerical results in Fig. 3. *We have added a relevant discussion to underscore this observation (highlighted in red color in Sect. 4.4, Lines 3-4 in the last paragraph of Page 7)*. Nevertheless, the main challenge lies within the existing processing method to annihilate the unused parameters. This assumption enables us to reconstruct the paired parameters into linear functions for further processing. When directly accommodating the ReLU basis functions, the current processing method faces significant challenges in estimating the $L^\infty$ error of ReLU basis function summation. In localized scenarios where only a subset of basis functions is summed up, there may exist a considerable $L^\infty$ error.
>    - As for canceling out the disparity between pairwisely and totally random scenarios with annihilate methods, we think that the existing approach is insufficient mainly due to its dependency on the pairwise assumption discussed previously. Nevertheless, upon increasing the network width, the totally random initialized weights will gradually approach those in pairwise initialization scenarios. Therefore, we believe the discrepancy between these two scenarios can be mitigated with additional permutation-based preprocessing strategies. We are endeavoring to address this issue.
>    - Actually, a strict control of the initialized weights is not required by the pairwisely random initialization, because the paired components are not required to appear adjacent to each other. Therefore, in practice, we can first randomly initialize a vector $W'$ with half of the length, then create a copy $-W'$ with reversed positive and negative signs, and finally concatenate the two vectors together as $W = [W', -W']$ (this is also how we implemented it in our programming).
> 2. **Divide into more steps to use up the remaining parameters.** We appreciate the reviewer's interesting suggestion. Unfortunately, upon careful consideration, we believe that the proposed approach would encounter substantial challenges, primarily due to the increasing difficulty of managing the positions and heights of the substeps. In fact, this approach is very similar to the *pseudo-copy* technique we introduced. However, according to the error analysis in App. H, we ascertain that this technique has controllable errors only if the substeps are close enough to the original step. However, the value of the remaining parameters is beyond our control. Consider the scenario of approximating a single step function at $s=0.1$ on the interval $[0,1]$ as an example. No matter how the step is subdivided, we cannot handle the parameters near the right end of the interval with a sufficiently small error.

---

> ### Author Response · Authors · 2023-11-20
> **Response to Reviewer BYmb (pt 2/2)**
>
> 3. **Connection between Fig. 4 and other topics.** We regret any confusion caused by the unclear explanation of Fig. 4, since the primary objective of this section is to report the observable phenomena. *To enhance comprehension, we have incorporated an additional subplot in Fig. 4 to show the behavior of loss reduction, and rewritten the relevant discussion.* An exhaustive and comprehensive interpretation of Fig. 4 necessitates additional experiments and theoretical investigation. We plan to further investigate this domain in future work. Here we discuss more explicitly the possible correlation between the observation and the topics raised by the reviewer as follows.
>    - **Rank structures of permutation groups.** Our study identified a thread-like pattern in the permutation training dynamic, implying a selective involvement of components in the permutation. This observation indicates that the training behavior can be fully described by the weights involved in the permutation. Therefore, certain low-dimensional structures may exist within the permutation training dynamics. This dimensional reduction could significantly alleviate the computational challenges of permutation search. Furthermore, exploring the potential mathematical depiction of these structures, such as the low-rank decomposition or approximation of the permutation group, could provide insight into how the network encodes information through training.
>    - **weight consolidation/pruning.** The idea of low-dimensional structures referenced above can also be naturally applied to the research of continual learning and the lottery ticket hypothesis. These domains require the identification of weights that are more critical to network performance, and our previously mentioned inherent structures offer a potential approach. Therefore, we believe that an avenue worth exploring is the application of permutation training to weight freezing and pruning. For instance, we can freeze certain weights associated with the inherent structures to retain the performance on past tasks. The pruning strategy can also be guided by referring to the level of activity in which weights participate in the permutation.
>    - **weight projection.** Regarding the weight projection method highlighted by the reviewer, it actually has little association with the observation in Fig. 4, and primarily correlates with the current algorithm implementation of permutation training. The LaPerm algorithm, introduced by Qiu & Suda (2020), leverages inner loops of free training to navigate weight permutations. Formally, it can be regarded as applying a value-preserving projection to the free training outcomes, and thus parallels the weight projection method derived from continual learning, such as OWM (Zeng et al. 2019). In the realm of continual learning, the projection ensures that the learned weights reside within the subspace of the preceding tasks. Consequently, we believe that these two domains can draw on insights from each other, leading to the development of more efficient training algorithms.

---

> > ### Comment · Reviewer_BYmb · 2023-11-23
> >
> > Thank you for the detailed answers. In light of the difficulty of handling true random initialization and multidimensional output I will keep my score. As discussed by other reviewers, I think it would be interesting to understand why some random initializations are more likely to allow UAP than others.

---

### Official Review · Reviewer_S6yU · 2023-11-01

**Soundness:** 3 good
**Presentation:** 3 good
**Contribution:** 4 excellent
**Rating:** 8
**Confidence:** 3

**Summary:**

This paper investigates the universal approximation property (UAP) of neural networks, specifically focusing on permutation-based training methods. The authors demonstrate that, without altering the exact values of neural network weights and only by permuting them, one can achieve effective approximation results, particularly for one-dimensional continuous functions. The research offers both theoretical proofs and empirical results, highlighting the potential of permutation training in shedding light on detailed network learning behaviors and its implications in various application scenarios.

**Strengths:**

*  The paper delves into the universal approximation property of permutation-trained networks, discussing the impact of various initialization strategies. A significant strength of the paper is the theoretical proof that a permutation-trained ReLU network can approximate one-dimensional continuous functions. The paper does not just rely on theoretical claims; it also presents numerical results which validate the performance of the permutation training method on regression tasks.
* The paper suggests that permutation training can serve as a novel tool for understanding network learning behavior. This could provide a fresh perspective on how neural networks learn and adapt.
* The observations made during permutation training are tied to other practical and important topics such as neural network pruning and continual learning.

**Weaknesses:**

* The exploration of practical aspects of permutation-based training, in particularly its potential applications to weight consolidation for continual learning and pruning, seem highly interesting and promising. Further elaboration on how this method supports such applications, potentially complemented by dedicated empirical evaluations, would greatly enhance the value of the proposed theory and the related version of the training method.
* Sparse training is of particular interest for its potential computational efficiency in resource-constrained environmental. An exploration of how the permutation-based method performs under sparse training conditions, such as with a random subset of weights initialized to zero, would provide valuable insights.
* The manuscript offers a valuable theoretical analysis with ReLU activation functions. It would be beneficial to investigate the extent to which these theoretical findings can be generalized to other activation functions, like leaky ReLU (or non-differentiable activation functions).
* The appendix, particularly Appendix A, contains details that may be of significant interest to readers, especially regarding hardware implementation benefits. Incorporating a summary of these details into the main text could reinforce the practical significance of the findings.

**Questions:**

* How is the permutation period k chosen in the relaxed version of the permutation-base training method? Table 2 lists the values used in the experiments but does not provide an intuition how the parameters were chosen.
* Fig. 1.c-d) are not sufficiently explained in the text. Could the authors offer a more detailed examination of the training dynamics illustrated in Figure 4? The current explanation offers a broad overview; however, the intricacies, such as why the permutations appear thread-like, remain unclear. A deeper analysis would be beneficial in understanding these nuances.

---

> ### Author Response · Authors · 2023-11-20
> **Response to Reviewer S6yU (pt 1/2)**
>
> We thank the reviewer for the supportive comments. We are encouraged that the reviewer found our work valuable and interesting. We hope to have addressed all the issues by the reviewer in our revised manuscript and in our replies below.
>
> 1. **Choice of permutation period $k$.** We acknowledge this insightful suggestion. As a hyperparameter, the choice of permutation period $k$ has the possibility to affect the convergence behavior. Qiu & Suda (2020) reported an unambiguous correlation between the value of $k$ and the final accuracy (refer to Fig. 6 in Qiu & Suda (2020)). Generally, a larger $k$ is associated with a marginal enhancement in accuracy for a single training process. However, the convergence behavior of our averaged results suggests that this effect remains negligible. *We have incorporated a new figure (Fig. 8 in App. N) aimed at evaluating the impact of $k$'s value on convergence behavior, and also added a relevant discussion in Lines 4-6 of the last paragraph on Page 6.*
>
> 2. **Fig. 1.c-d) are not sufficiently explained.** We apologize if these were not clearly explicated within the initial draft.
>
>    - Fig. 1(c) plots the step function approximator $f_{s}^{\text{NN}}(x)$ as light green color, which is aimed at approximating a given step function $f_s(x)$ in orange color. The approximator is constructed by four-pair basis functions $\phi_i^\pm$, $i = 1,\cdots,4$, from Eq. (3), where $b_i$ $i = 1,\cdots,4$, note the location of basis functions. The distance between the adjoint basis functions $b_1$ and $b_2$ is noted as $d$ (i.e., $d = b_2 - b_1$), and the symmetry condition ensures that $d = b_4 - b_3$.
>    - Fig. 1(d) illustrates the idea of the pseudo-copy technique, which is proposed to get rid of the scaling factor $\alpha$ and $\gamma$ in Eq. (6). The resulting function $\acute{f}_s^{\text{NN}}$ is plotted as a dark green color, which is summed by the pseudo-copies in light green color. Each pseudo-copy shares the same form with the approximator plotted in Fig. 1(c).
>
>    *We have promptly addressed this oversight in our revised version by providing more descriptions (highlighted in red in the figure capture of Fig. 1) and recall for both figures.*
>
> 3. **More detailed discussion of Fig. 4.** We appreciate the reviewer's interest and regret any confusion caused by the unclear explanation of Fig. 4, since the primary objective of this section is only to present the observable phenomena. *To enhance comprehension, we have incorporated an additional subplot in Fig. 4 to show the behavior of loss reduction, and rewritten the relevant discussion.* By comparing the loss behavior, it becomes evident that the decrease in loss in the third stage—characterized by the appearance of a thread-like pattern—is noticeably slow. Therefore, a rather slow update of certain parameters fails to trigger significant shifts in the order relationship. Moreover, the localized permutation behavior was also reported by Qiu & Suda (2020), which indicates that this behavior may not be confined to the third stage (as plotted in Fig. 14 in the Appendix of Qiu & Suda (2020)). However, an exhaustive and comprehensive interpretation of Fig. 4 necessitates additional experiments and theoretical investigation. We plan to further address this issue in future work.

---

> ### Author Response · Authors · 2023-11-20
> **Response to Reviewer S6yU (pt 2/2)**
>
> 4. **Sparse training.** We agree that this is a scenario worth further exploring, since Qiu & Suda (2020) have also reported the significant advantages of permutation training in sparsity scenarios (see Fig. 6(e) in Qiu & Suda (2020)). In our future research, we aim to depict this immense potential theoretically. On an intuitive level, our proof can explain the advantages of permutation training in sparsity scenarios. Notice that a primary challenge of permutation training is managing unused basis functions. However, in sparse situations, these remaining basis functions can be conveniently dealt with by assigning them a zero coefficient, thereby facilitating the approximation process. *We have added the relevant discussion to the App. P of the revised version.*
>
> 5. **Generalization to leaky-ReLU activation functions.** We believe that extending our outcomes to leaky-ReLU can be straightforward. *Our newly added numerical experiment also demonstrates the approximation power (see Fig. 10 in App. O)*. This is because of the two crucial techniques deployed in our proof, specifically, constructing the step function approximators and eliminating the unused parameters, both of which can be readily applied to the leaky-ReLU.
>
>    - It's feasible to construct a similar step function approximator by rederiving the relevant coefficient $p_i, q_i$, since our two-direction setting of basis function (i.e., $\phi^+$ and $\phi^-$) can impart leaky-ReLU with symmetry equivalent to ReLU.
>    - Furthermore, the existing eliminating method can be directly employed for leaky-ReLU, due to the capacity to reorganize a pair of leaky-ReLU basis functions into linear functions for further processing.
>
>    *We have added the relevant discussion to the revised version (highlighted in red in Lines 1-2 of the first paragraph of Sect. 4.4 on Page 7).* However, given the limitation in paper length, the majority of corresponding contents are organized in the App. O.
>
> 6. **Incorporating a summary of appendix into the maintext.** We appreciate the reviewer's interest and *have positioned relevant summaries within Sect. 1 of the revised version (highlighted in red in Lines 3-6 of the second paragraph of Page 1)*. However, given the limitation in the paper length of the final version, we are afraid that the majority of corresponding discussions will likely be situated in the appendix.

---

> > ### Comment · Reviewer_S6yU · 2023-11-21
> >
> > I read the authors' responses and appreciate the effort they put into addressing my concerns.

---

### Meta-Review · Area_Chair_HsjC · 2023-12-06

**Metareview:**

This paper shows that neural networks with ReLU activation can approximate arbitrary continuous functions under parameter replacement. This study is motivated by the recent replacement-based learning method. The content is excellent in characterizing the advantages of the new learning method. The paper is written in a clear and readable manner. There are criticisms that the setting is limited to two-layer neural nets and one-dimensional settings, but given that this research topic is still emerging, these limitations do not detract significantly from the contribution of the paper.

After careful reading, my only concern is that the results of this paper only show the existence of replacements that achieve universal approximation, and do not discuss how to discover these replacements in the study. In practice, the number of possible replacements grows exponentially, so it is not easy to discover the replacements that do exist. In other words, this paper is not based on replacement training method, but rather a modification of the usual universal approximation theorem with restrictions on the realized value of parameters. The theoretical contribution is even more limited when one takes into account that the realizations are also obtained from very fine equally spaced points.

Taking into account these shortcomings and the fact that the reviewers did not agree on the adoption of the paper, we made this decision.

**Justification For Why Not Higher Score:**

I had thought that this paper should be accepted, but I have come to believe that it is only a minor modification of the usual universal approximation theorem, since it does not present a method for learning replacements to achieve the approximation.

**Justification For Why Not Lower Score:**

N/A

---

### Decision · Program_Chairs · 2024-01-16

Reject